# Recent Advances in the Rejection of Endocrine-Disrupting Compounds from Water Using Membrane and Membrane Bioreactor Technologies: A Review

**DOI:** 10.3390/polym13030392

**Published:** 2021-01-27

**Authors:** Kamil Kayode Katibi, Khairul Faezah Yunos, Hasfalina Che Man, Ahmad Zaharin Aris, Mohd Zuhair bin Mohd Nor, Rabaah Syahidah binti Azis

**Affiliations:** 1Department of Agricultural and Biological Engineering, Faculty of Engineering and Technology, Kwara State University, Malete 23431, Nigeria; kamil.katibi@kwasu.edu.ng; 2Department of Biological and Agricultural Engineering, Faculty of Engineering, Universiti Putra Malaysia, UPM Serdang 43400, Selangor, Malaysia; hasfalina@upm.edu.my; 3Department of Food and Process Engineering, Faculty of Engineering, Universiti Putra Malaysia, UPM Serdang 43400, Selangor, Malaysia; zuhair@upm.edu.my; 4Department of Environment, Faculty of Forestry and Environment, Universiti Putra Malaysia, UPM Serdang 43400, Selangor, Malaysia; zaharin@upm.edu.my; 5Material Processing and Technology Laboratory (MPTL), Institute of Advance Technology (ITMA), Universiti Putra Malaysia, UPM Serdang 43400, Selangor, Malaysia; 6Department of Physics, Faculty of Science, Universiti Putra Malaysia, UPM Serdang 43400, Selangor, Malaysia; rabaah@upm.edu.my; 7Materials Synthesis and Characterization Laboratory (MSCL), Institute of Advanced Technology (ITMA), Universiti Putra Malaysia, UPM Serdang 43400, Selangor, Malaysia

**Keywords:** endocrine disrupting compounds, occurrences, membrane processes, removal mechanisms, membrane bioreactor (MBR) process, fouling mitigation

## Abstract

Water is a critical resource necessary for life to be sustained, and its availability should be secured, appropriated, and easily obtainable. The continual detection of endocrine-disrupting chemicals (EDCs) (ng/L or µg/L) in water and wastewater has attracted critical concerns among the regulatory authorities and general public, due to its associated public health, ecological risks, and a threat to global water quality. Presently, there is a lack of stringent discharge standards regulating the emerging multiclass contaminants to obviate its possible undesirable impacts. The conventional treatment processes have reportedly ineffectual in eliminating the persistent EDCs pollutants, necessitating the researchers to develop alternative treatment methods. Occurrences of the EDCs and the attributed effects on humans and the environment are adequately reviewed. It indicated that comprehensive information on the recent advances in the rejection of EDCs via a novel membrane and membrane bioreactor (MBR) treatment techniques are still lacking. This paper critically studies and reports on recent advances in the membrane and MBR treatment methods for removing EDCs, fouling challenges, and its mitigation strategies. The removal mechanisms and the operating factors influencing the EDCs remediation were also examined. Membranes and MBR approaches have proven successful and viable to eliminate various EDCs contaminants.

## 1. Introduction

The continuous expansion in the global population and expeditious industrial progression resulting in the production of emerging chemicals poses a severe threat to access to quality and clean water. Over the past 20 years, awareness of the presence of dangerous and intransigent contaminants, generally known as endocrine-disrupting compounds (EDCs), has risen in the terrestrial and aquatic ecosystems setting [1]. The noticeable and incessant occurrence of endocrine-disrupting compounds (EDCs) as emerging environmental pollutants has led to considerable interest among the researchers during the past few years owing to their potential human and environmental risks [2]. World Health Organization (WHO) describes an endocrine disruptor as “an exogenous substance or mixture that alters function (s) of the endocrine system and consequently causes adverse health effects in an intact organism, or its progeny, or (sub) populations” [3]. The most critical global concern is presently centered on the persistent occurrence of endocrine-disrupting chemicals (EDCs) and other synthetic chemicals in the environment. This issue is further amplified due to the increased expansion in the chemical release, which has now attained more than 400 million tons worldwide, and the increased pollution from these chemicals. As such, the impact on human health through known or unknown effects of these chemicals on hormonal systems is pronounced [4].

Conventional wastewater treatment systems are inadequate to properly remove EDCs contaminants from urban wastewater and consequently discharge untreated pollutants into downstream water supplies, which pose a severe threat to humans and the environment [5]. However, rapid and substantial advances in wastewater treatment have been made to address the issue of water contamination regarding EDCs and have demonstrated to improve their removal, including adsorption, advanced oxidation techniques, and bioremediation, among others [6,7,8,9,10]. However, these approaches are unattractive, and their applications have been seriously constrained owing to numerous limitations, such as high energy demand, substantial operational costs. Moreover, the use of austere environmental conditions, secondary sludge disposal challenge, a requirement of toxic chemicals that are capable of producing toxic and persistent transformational by-products even in some circumstances are more than the initial constituents compound [11]. Interestingly, one avenue to address this challenge is to consider a novel physical, efficient, non-toxic, and low-cost approach. Membrane technology is a phase-changing approach and a promising option to conventional treatment procedures. Membranes can be fabricated from various materials that provide them with particular physicochemical properties and the capacity to successfully eliminate a broad spectrum of endocrine disruptors [12]. To the best of our knowledge, a dedicated review of the rejection of endocrine-disrupting compounds via membrane and membrane bioreactor (MBR) is lacking.

The focus of this paper is to provide a critical review of the rejection of endocrine-disrupting compounds via membrane and MBR technologies. Accordingly, the occurrence and sources of EDCs as contaminants and their adverse impacts on the environment would be introduced and discussed. The membrane removal mechanisms, factors controlling EDCs rejection, membrane fouling, and abatement strategies, including membrane modification techniques, would also be reviewed and discussed. Finally, the main conclusions and future perspectives were presented.

### 1.1. Occurrences of EDCs as Contaminants and Its Sources

EDCs typically comprise of natural estrogens and synthetic substances engineered nanomaterials, pesticides, pharmaceuticals, personal care products, drugs of abuse, as well as other industrial chemicals with a high propensity to stimulate estrogenic effects, harmful impacts on the endocrine systems of humans, fauna, and available water resources [13,14]. Though there exist numerous multiclass compounds categorized as endocrine-disrupting compounds, natural estrogens including estrone (E1), estradiol (E2) and estriol (E3), synthetic estrogen bisphenol A (BPA), 17-α-ethynylestradiol (EE2), and nonylphenol (NP) have received massive global interest (see Table 2) amongst many EDCs owing to their harmful consequences on public health and the environment [15,16].

Steroidal estrogens, such as estrone (E1), estradiol (E2), together with estriol (E3), are naturally occurring in animal and human bodies through 17-α-ethynylestradiol (EE2) is produced primarily for contraceptive pill [17]. Notably, compounds, such as 17β estradiol (E2), 17α-ethinyl estradiol (EE2) together with estriol (E3), are currently obtained in the watch list of EU Commission Decision 495/2015 and are considerably investigated, due to their elevated estrogenicity at minuscule proportions (µg/L and ng/L) and their detection in various environmental matrices particularly effluent from sewage treatment plants, surface, drinking, and groundwater (see Table 1) [18,19].

Also, environmental xenoestrogens, including bisphenol A (BPA), which flow into WWTPs via discharges from the industrial sector and leaking from BPA-based products [20], 4-tert-octylphenol (4-t-OP), and 4-nonylphenol (4-NP) are synthesized with increased volume of production [21].

There is no doubt that EDCs contaminants are typically detected in the range of nanograms to micrograms (ng/L and μg/L) in the environment and predominantly occur in numerous matrices, for example, soils, water (groundwater, wastewater, surface waters, drinking water), biota, sediments, and air [17,22,23,24], conceivably triggering possible hazards to public health and the ecosystems as presented in Table 2. As indicated in Figure 1, these contaminants can infiltrate directly into the aquatic environment via effluent outflow and indirectly as run-off. Nevertheless, the primary route of EDCs contaminants to the freshwater bodies is treated and raw urban effluent release into waters-bodies [25,26]. Moreover, most of the treated potable-water resources may be polluted through deep-well injection of the effluent and surface outflow [27]. This flaw provides evidence that even drinking water is not free from these recalcitrant contaminants, since some compounds of EDCs, specifically plasticizers and steroidal hormones, were detected in drinking water, surface water, and groundwater (see Table 1) [28,29,30,31].

Similarly, studies from Jonker et al. [32] and Cai et al. [33] in their studies reported that EDCs in the water bodies emanate from human-induced sources (anthropogenic), namely, industrial wastewaters and effluent of municipal, run-off water from polluted soils via pesticides containing EDCs compounds, such as alkylphenol polyethoxylates (APnEOs) or alkylphenols (APS), and the application of sludges (sewage) on cultivated fields. ApnEOs and APs have a propensity to accumulate and segregate in the environmental sediment. Based on bio-surveillance data, the public at large is susceptible to ApnEOs and APs [34,35]. However, contaminated food and drinking water are the leading cause of human exposure [36,37] or via interaction with detergents and personal care products [29]. This anomaly is due to a substantial proportion of drugs consumed by patients, thereby penetrating through their body unchanged and traverse through human excrement to the wastewater [38].

Notably, the principal route of phenolic EDCs, including BPA and its analogous in an intact organism, is through ingestion, thereby recording almost 90% of BPA vulnerability [39]. The sludge generated in WWTPs is frequently applied in cultivated fields as a soil improvement, in which the detection of these compounds in soils with low sorption affinity may further contaminate the nearby surface and ground waters via infiltration and run-off [40].

Specifically, the pollution of subsurface or groundwater with persistent EDCs microcontaminants is majorly resulting from an interaction between surface and groundwater via soil, sewer systems, landfill leachate, percolation of polluted water from agricultural lands, and seepage of septic systems.

The EDCs contaminants can be analyzed using chemical analytical techniques in aqueous matrices, including high-performance liquid chromatography with mass spectroscopy, gas chromatography with mass spectroscopy, and biological approaches enzyme-linked immune-sorbent assay. Besides, there is increasing utilization of biosensors for this function [41,42]. Various sources and pathways of endocrine-disrupting compounds in the environment are illustrated in Figure 1.

### 1.2. Adverse Effects of EDCs on the Environment

Numerous pieces of literature have reported various deleterious impacts of EDCs on the environment and its propensity to distort stability in the ecosystem. Table 1 summarizes some of the harmful effects of frequently discovered EDCs pollutants together with their corresponding concentrations in drinking water, surface water, and wastewater. Bitty concentrations of these emerging pollutants (ng/L to µg/L) have been proven to significantly induce severe consequences on the ecosystem and the health of vertebrate species [44].

Generally, contamination of water, due to the presence of EDCs contaminants, causes several detrimental impacts on both public and wild animals [45]. These chemicals could interfere with the hormonal functions in the endocrine system by functioning as hormone mimics, modifying the metabolism and synthesis of natural hormones, or altering hormone sensory receptor levels and receptor agonists/antagonists [46]. EDCs can presumably lead to many health problems [47]. For instance, the most detrimental and threatening influence occurs in their tendency to trigger reproductive disorders in various species, including humans. The estrogenic impact of EDCs is often described in terms of estradiol equivalent (EEQ), and it has been established that a concentration of 1 ng L^−1^ EEQ has a severe effect on fish and other marines [48]. EDCs can feminize male fish, adversely impact reproductive performance, decreased sperm counts, and elicit fastllogenin formation [49].

Plasticizer EDCs, such as bisphenol A (BPA) and 4-n-nonylphenol (4-NP), are reduced by several orders of magnitude through estrogenic activities, but are often notable for their elevated levels in drinking water, treated wastewater effluent, surface sediment, and aquatic organism (fish) (see Table 1) [50]. EDCs have also been linked to altered behavior and obesity in children, reduced gonadal development and viability, and altered humans and wildlife [51,52]. The consequences of these persistent contaminants are not only restricted to an adult individual, but the possible mechanism of transgenerational epigenetic inheritance can also be transferred on to future generations [53].

The persistence of EDCs in water, even at trace concentrations ranging from little ng/L to numerous µg/L, are sufficient to elicit endocrine disruption in many species. It is notably dangerous to health, due to its ability to trigger metabolic and reproductive disorders; hence, the need for efficient management of EDCs contained in effluent before discharge is indispensable [54]. Due to environmental hazards linked to the EDCs contaminants, the treatment of outflow discharges emanating from various sources, such as pharmaceutical compounds, pesticides, personal care products, and similar compounds, have received significant global attention [55]. More stringent standards are still necessary for efficient control of these recalcitrant micropollutants, thereby providing considerable control [43]. Primarily, the most practiced management technique is a conventional treatment.

Globally, several EDCs contaminants have been detected in effluent discharged from WWTF, frequently at a proportion of hundreds of nanograms per liter to micrograms-per-liter scale [40]. Furthermore, there has been an increasing amount of literature that emphasized and reported series of health challenges associated with endocrine-disrupting compounds, such as interference with the endocrine system of man and animals by influencing the synthesis, release, transport, metabolism, and excretion of hormones in the body, mimicking, blocking, disrupting the normal function of hormone system in humans thereby causing severe effects, such as abnormal reproductive growth, cardiovascular changes, reduction of sperm reproduction in humans which result to low fertility, thyroid and adrenal gland dysfunctions, immune, neurological diseases, developmental dysfunctions throughout the fetal period, stimulation of breast cancer in women, development of testicular and prostate cancer, a decline in reproductive fitness of men and increased threat to human [19,56,57,58,59,60,61,62,63,64,65]. In addition, EDCs have also be linked with altered behavior and obesity in children, reduced gonadal development and viability, alter physiological status in humans and wildlife [51,52].

Mostly, exposure to EDCs by humans and animals is through ingestion of accumulation and biomagnification concerning species at a high level of the food chain [66]. Furthermore, research findings also reported negative impacts of EDCs towards animals as it affects the hormonal systems of organisms, inhibiting regular action of the endocrine system, binding to estrogen receptors in wildlife, and mimic the actions of endogenous estrogen, causing reproductive disorders, feminization, and carcinogenesis in numerous wildlife animals, interfering with the synthesis release, transport, combine and interact with female estrogen and disturb the reproduction, growth, and behavior of organisms, interfere with a delicate balance of the endocrine system of animals, alter the normal hormone functions and physiological status in wildlife and threat to health and reproductive biology in an animal population [51,52,65,67,68,69,70,71].

Also, previous studies have widely reported various anomaly observed in the aquatic environment as a result of the presence of EDCs which includes: Bioaccumulation and biomagnification in the marine ecosystem, intersex and skewed sex ratios, reduction in fish fertility, abnormal blood hormone levels, altered gonadal development (imposex and intersex), induction of vitellogenin gene and protein expression in juveniles and males, masculinization/feminization, disruption of the reproductive mating behavior of fish, intersex in white suckers fish downstream of a wastewater treatment plant effluent, hermaphroditism, decreased fertility and fecundity [67,69,71,72,73]. Exposure to EDCs has also been reported to pose a potential risk on the water quality and the ecosystem because EDCs can present a potential risk to the ecosystem, affects water quality, increase adverse ecological impacts, and be considered as an environmental pollutant with comparatively high biological activity [62,64,74].

Continuous consumption and disposal of EDCs into urban sewage have consequently caused the conventional treatment systems to be a possible and important route of EDCs contaminants in the vicinity. Hence, the release of treated water from traditional treatment facilities into groundwater, open water, and other waterbodies regularly can efficiently heighten the tenacity of EDCs in the vicinity, since findings have revealed that minuscule amounts of major EDCs are eliminated from the conventional treatment systems [75,76]. In some instances, the concentration of EDCs effluent could surpass the feed concentration, mainly owing to the biological changes taking place during biological degradation [77].

Furthermore, the findings have shown that in particular, EDCs can alter endocrine functioning by harming the normal physiological reactions concerning the male and female reproductive orderliness (such as menstrual cycle abnormalities, alteration of hormone concentration, impulsive abortion, endometriosis, and polycystic ovarian disorder [78,79,80].

For instance, Giulivo et al. outlined the possible role of EDCs (such as phthalates, bisphenol A and parabens) on the pathogenesis of breast cancer at minimal proportions [81].

The impact of chronic and acute vulnerability on the reproductive function, histopathological variations, and body organs of fishes, birds, mammals, and mud snails has also been described elsewhere [82,83,84]. Desai et al. also elucidated the effects of EDCs on metabolic syndromes, including obesity, cardiovascular disorders, insulin resistance, dyslipidaemia, hepatic damage, and type2 diabetes in individuals [85].

The challenge to be tackled is how there can be a drop in the input of the EDCs in the environment. Kummerer [86] suggests that the technical approach of improving the conventional treatment systems to advanced treatment applications as a short-term to medium-term strategy along with the replacement of dangerous compounds used in the manufacturing of chemical compounds with more non-threatening chemicals as a sustainable strategy will be a beneficial means for managing the associated risk of EDCs in the environment. Hence, conventional treatment systems should be upgraded with advanced treatment technologies, including membrane separation techniques, membrane bioreactor (MBR), and other advanced oxidation processes, ultraviolent irradiation to forestall the challenges associate with the extermination of these persistent contaminants.

## 2. Rejection of EDCs by Membranes

Membranes are permeable and thin material layers employed to eliminate water pollutants by allowing water to be conveyed at varying rates based on the membrane pore size. [87]. Membrane technology is the most widely applied technique for the elimination of microbes and salt from water. It serves as selective filters or screens, eliminating contaminants bigger than the pore size of the membrane and permitting small-sized contaminants and water molecules to permeate [88]. Membrane processes have been utilized in drinking water and wastewater reuse to remove EDCs and natural organic matter (NOM) [89]. Principally, the membrane is described mostly by driving pressure exerted to the microfiltration (MF), ultrafiltration (UF), nanofiltration (NF), and reverse osmosis (RO) separation process [90]. These processes, as shown in Table 1 and Table 2, comprising of comparatively low-pressure systems, particularly ultrafiltration (UF) and microfiltration (MF) operating at pressures ranging between (5–10 bar), respectively, or high-pressure systems, such as nanofiltration (NF), working at practically 50 bar and reverse osmosis (RO) up to 70 bar (or 150 bar for high-pressure RO systems) [91,92]. These systems have been employed in the rejection of EDCs organic contaminants from different water matrices. However, RO has more critical fouling challenges and displays excellent removal efficiency [93].

### 2.1. Membrane Materials Used in the Rejection of EDCs

Generally, the efficacy of critical components in the fabrication of membranes lies in the precise choice of materials and techniques. Different polymers have been periodically applied from various studies to remediate recalcitrant EDCs from potable water and wastewater [96]. Various membrane materials could be explored to reject endocrine-disrupting compounds from synthetic, real water, and sewage effluent discharges to provide potable drinking water. This is because the occurrence of these organic pollutants in potable water and wastewater must be adequately relegated before releasing it into any receiving water body to safeguard aquatic life and ecosystem. Membranes can be fabricated from different materials, offer them specific physicochemical properties, and the capacity to reject a broad spectrum of endocrine disruptors [12].

Notably, membrane materials possess the ability to influence their fouling susceptibility. Principally, membranes can be categorized into polymeric, ceramic, and composite membranes based on the membrane material. The mechanical strength, pore size distribution, porosity, durability, cost, polymer flexibility, fouling resistance, stability, wetting proneness, and chemical resistance are the critical factors to be considered during the material selection [97].

Ceramic membranes display excellent filtration efficiency thanks to their elevated chemical resistance, static nature, integrity, and facile cleaning resulting in minimal operational expenses [98,99]. Ceramic membranes are also extremely hydrophilic [100], which enhances their fouling tenacity. These membranes are one of the most frequently employed materials, specifically for the anaerobic membrane bioreactor (AnMBR) technique [101], owing to their improved control of fouling and concentration polarization via backwashing, along with efficient defiance to abrasion and corrosion [102]. Other materials are also required during their application in either flat sheet and hollow fiber design [103]. However, their substantial fabrication expenses and delicate nature [98] render them financially unsustainable for industrial sectors.

Polymeric membranes consist of a polymer monolith and remain the leading and broadly applied materials in membrane technique owing to numerous benefits of superior flexibility, excellent chemical and physical resistance, firmness, and separation characteristics, though usually hydrophobic [100]. In recent years, polymeric membranes have attracted increasing interest in both scientific and industrial applications. At present, practically all polymeric membrane materials consist of organic and inorganic polymers for engineering applications, but the latter control the global membrane sector. Typically, organic polymers include poly (vinylidene fluoride) (PVDF), polyacrylonitrile (PAN), polysulfone (PSF), polyethersulfone (PES), polyamide, polyimide, poly-(1,4-phenylene ether) ether sulfone (PPEES), poly (ether sulfone), polytetrafluoroethylene (PTFE), and poly (p-phenylene sulfide) (PPS), polyurethane (PU), polypropylene (PP) and cellulose acetate (CA). Polymeric membranes, due to their hydrophobic characteristics, have a propensity to foul facilely, yet they have extensively applied now thanks to the facile fabrication of the pore sizes. Generally, polymeric membranes exhibit properties of being a different material, self-supported, and the unique material used to construct hollow fiber membranes [104]. Besides, these membranes are low-cost for industrial applications [105].

Composite membranes are recently fabricated from multiple materials to merge the component materials’ strength in the resultant product. Typically, one material represents the active surface, and another makes the base layer [13]. In composite membrane applications, hydrophobic membranes are coated with a hydrophilic polymer to overcome the fouling issue. Table 3 presents the summary of findings regarding the use of various polymeric materials, their fabrication techniques, and their corresponding modules, recently reported in some literature. As shown in Table 3, PVDF, PSF, PES, CA, and PAN are employed in various membrane fabrications. Of all these, PVDF is the most preferred and widely applied polymeric membranes and has attracted increasing attention in recent years from researchers and producers. The use of PVDF polymer has shown promising and unique properties that present it as a better candidate to eliminate EDCs contaminants from water. These properties include excellent thermal stability and mechanical strength, good chemical resistance, and exceptional aging resistance, crucial for the practical application of membrane technology [106]. Besides, PVDF displays satisfactory processability to fabricate hollow fiber (HF), flat sheet, and tubular membranes and dissolvable in several conventional solvents, including dimethylformamide (DMF), *N*-methyl-2-pyrrolidone (NMP), and *N*, *N*-dimethyl acetamide (DMAc) [107]. As regards membrane processes, UF and MF membranes principally employ polymers, for example, polyethersulfone (PES), polysulfone (PSF), polyacrylonitrile (PAN), polyvinylidene fluoride (PVDF), polytetrafluoroethylene (PTFE), and polypropylene (PP) as membrane constituents. These materials demonstrate outstanding selectivity, permeability, and stability (mechanical, thermal, and chemical) for water purification purposes. Comparatively, PES and PSF membranes appear to be in the midst of increasingly used materials for UF applications. Undeniably, these standardized polymers are also utilized in the fabrication of RO and NF hybrid membranes, whereas PVDF and PP are more distinctive for MF membranes [108].

Nevertheless, it is necessary to optimize and improve the efficacy of separation of these polymeric membranes [109] and enhance specific other physical characteristics, for instance, fouling resistance, stability, and hydrophilicity profile [110]. Significantly, membrane polymers have been improved by the accretion of supplementary materials or blending with several fillers to improve the resulting membranes. Besides, the formulation of membranes is influenced by essential morphology.

Regarding membrane-fabrication techniques, numerous approaches are applied to fabricate membranes comprising phase-inversion, interfacial polymerization, electrospinning, stretching, and track-etching techniques [111]. These techniques may probably rely extensively on the material (membrane) compared to membrane processes (for example, MF and UF), while processes differ in terms of pore size, particularly for composite membranes. Correspondingly, as indicated in Table 3, electrospinning and phase inversion are the most frequently applied methods to manufacture PVDF, PSF, CA membranes predominantly exploited for the rejection of EDCs from potable and synthetic water. Technically, the phase inversion method comprises non-solvent induced phase separation (NIPS) (a leading method), thermal phase separation, and regulated evaporation, and lastly, vapor induced phase inversion (VIPS) [112]. Most of the composite membranes utilized in the rejection of EDCs contaminants possess pore size ranged from 0.0003–357 µm, which is comparatively lower than the size of some EDCs contaminants, therefore might be efficient to retain some of the EDCs contaminants during the membrane filtration processes as indicated in Table 3.

### 2.2. Removal Mechanisms of Endocrine-Disrupting Compounds during Membrane Processes

Generally, knowledge of the particular mechanisms accountable for removing endocrine-disrupting compounds is highly essential better to comprehend the separation of EDCs via membrane separation procedures.

In the membrane separation processes, the scientific literature extensively agrees with three fundamental removal mechanisms responsible for removing EDCs, namely, steric hindrance, adsorption onto the membrane, and electrostatic interactions [113,114]. However, the removal mechanisms vary for different membrane processes (MF, UF and NF, RO), respectively. The efficiency of these mechanisms on EDCs removal principally depends on some pertinent factors, including membrane type, the occurrence of pores and structure or morphology, characteristics, fouling, working parameters, and specific EDCs pollutant physicochemical properties [59,115], whereas hydrophobicity (often represented by dissociation constant (pK_a_) and octanol/water partition coefficient (LogK_ow_), and molecular geometry (such as width and weight) are the most influential parameters [116]. The EDCs removal mechanisms are illustrated (Figure 2).

#### 2.2.1. Removal Mechanisms of Endocrine-Disrupting Compounds during MF and UF Processes

Principally, the mechanism of removal in microfiltration and ultrafiltration membranes is via adsorption [117]. Adsorptive removal occurs when EDCs pollutants are adsorbed towards the membranes [118]. Studies of EDCs contaminants removal by the membrane filtration process reveal that the adsorption of EDCs not only takes place on the membrane surface, but in the pores, membrane skin layers, and the support layers [119]; however, it is hard to differentiate the influences of various components of membranes to sorption. Thus, this mechanism (hydrophobic interactions) plays a significant contribution in the rejection of microcontaminants, most especially for the adsorption of hydrophobic compounds onto membrane during membrane processes, as demonstrated in (Figure 3) [120].

The adsorption mechanism may be characterized by either one or by combining the three stages: (i) Diffusion of EDCs to a sorption site in membrane whether with pore diffusion method via solid surface or by a liquid-filled pores mechanism, (ii) mass transfer (liquid to particle surface phase) throughout the interface, and (iii) adsorption towards the membrane depending on the bonding operation (chemical/physical) [121]. Moreover, the adsorption of microcontaminants takes place through hydrophobic interactions (physical) and ionic, hydrogen, or covalent bonding (chemical) processes, as demonstrated in (Figure 2). Correspondingly, Okugbe et al. stated that the adsorption mechanism is via hydrophobic interaction and hydrogen bonding (chemical) between EDCs and membrane medium (physical) for the rejection of BPA in an aqueous environment [122].

Notably, several studies have reported that adsorption is the principal mechanism in UF and MF membranes for the separation of EDCs microcontaminants from water/wastewater as there exist sufficient available adsorptive sites in microporous membranes [123,124]. The tiny pores may influence the interaction of EDCs contaminants with various membranes layers; for instance, Comerton et al. assessed the sorption of EDCs contaminants using UF/NF/RO membranes and found that bigger pore sizes could enable more contact to the membrane’s inner sorption sites, therefore expanding the sorption of EDCs contaminants [125]. The authors also reported that the maximum adsorptive removal of BPA was attained with the UF during membrane filtration, suggesting that solute adsorption could occur both towards the membrane surface and membrane pores. Therefore, the membrane with wider pore sizes, particularly UF, poses better sorption sites regarding microcontaminant removal.

The initial elimination of EDCs contaminants by adsorption is often very high before a saturation point whereby the compound’s elimination could occur. The compound adsorbed may be dissolved across the membranes’ uppermost surface before moving across the membrane through diffusion or convection. The removal rate of EDCs contaminant could be undermined when the feed compound concentration is less than the equilibrium level, allowing the compound to desorb into the permeate portion. Similar studies by Kiso et al. have pointed out that higher removal of most compounds with higher hydrophobicity during NF membrane filtration is due to the stronger affinity between the solute and the membrane as demonstrated via octanol-water partition coefficient (K_ow_) [126]. Van Der Bruggen et al. undertake a study to relate molecular adsorption parameters comprising polarizability, dielectric constant, dipole moment, K_ow_, and molecular size [127]. They concluded that K_ow_ and molecular size are the two critical key parameters to elucidate the adsorption of hydrophobic compounds to membranes. Figure 3 demonstrates the adsorption interaction in the membrane via covalent, ionic, hydrogen bonds, and hydrophobic interaction.

Since MF and UF membrane pore sizes are usually much bigger than the molecular size of EDCs; thus, permeation via sieving mechanism does not influence removing EDCs. Nevertheless, rejection of BPA via size screening/exclusion of UF membrane is plausible to take place in the fouled membrane following the build-up of the cake layer, which forms a further impediment to retain BPA. This is corroborated by a comparative study from Hu et al. that stated that clean membrane could only reject BPA via adsorption, whereas the fouled membrane was efficient to reject BPA by a combined effect of adsorption and size screening interaction [129]. Findings revealed that the fouled membrane could remove 64–76% of BPA as compared to only 34% recorded in the clean membrane. Similarly, the rejection of EDCs contaminants by UF via electrostatic interaction could be desirable, since the separation of charged solutes via electrostatic interaction commensurate with the surface charge density for MF and UF membranes [130]. For instance, Shao et al. reported that adequate membrane (UF) charge alteration could strengthen anti-fouling properties and eliminate natural organic matter (NOM) as compared to unmodified UF membrane [131]. Findings indicate that adsorption rate declines with time once approaching equilibrium after the starting periods of separation; also, adsorption impacts quantified at the initial periods of separation present higher rejection nevertheless could be an overrating of the rejection, since once membrane reaches equilibrium point, removal diminishes drastically [125,132].

#### 2.2.2. Removal Mechanisms of Endocrine-Disrupting Compounds during NF and RO Membrane Processes

Steric hindrance (size screening) and electrostatic interaction describe the mechanisms of removal of EDCs in reverse osmosis and nanofiltration membranes, respectively [133,134]. Owing to the size screening/exclusion effects of a membrane, it can be anticipated that the performance of membrane filtration is strongly connected to the pore size [135]. Multiple mechanisms could occur, particularly in NF membrane owing to the porous and dense components, in addition to charges on their surface [115]. The characteristics of both the EDCs pollutants and membrane dictate solute-membrane interactions.

There are numerous indicators for assessing these properties, such as hydrophobicity, MWCO, charge, molecular size, diffusion coefficient, and among others. These parameters should be adequately considered when investigating the influence of the three main mechanisms of compound removal in NF [136]. The RO and NF membranes could adequately retain a vast majority of EDCs pollutants, dissolved solutes, and salts owing to their molecular weight in the range of 200–400 Da [137]. Comparatively, the removal efficiency of RO is higher than NF membranes, since the former can retain a broad array of EDCs pollutants, due to their smaller pores [138]. Nevertheless, NF membranes have other distinctive properties that favor their usage; for instance, the efficacy of retention is very close to RO membranes, with the prospect of working with more substantial flows and lesser pressures, minor fouling proportions, and lower expenses [138].

Steric hindrance (size exclusion) refers to the physical hindering of bigger solutes that exceed the membrane pore size by hindering the solutes from navigating via the membrane (see Figure 3) [118]. Generally, the sieving effect occurs in reverse osmosis and nanofiltration membranes, due to their pore size in nm-scale [139]. Numerous studies have reported that the molecular size of target compounds is the most critical parameter in assessing the removal rate of EDCs in NF [140,141,142,143,144,145]. A recent study from Xu and co-workers [146] found that potent steric hindrance interaction was strongly responsible for significant removals (100% pharmaceuticals) for candesartan and bezafibrate. Predictably, hydrophilic compounds enlarge their molecular sizes by developing a comparatively high affinity towards the water phase, and consequently, a sizeable hydrated radius. Compounds with high hydrophilicity mostly have a relatively considerable number of O or OH groups that can develop hydrogen bonds with water molecules and increase the efficient diameter [147,148]. Size exclusion mechanism is frequently noticed with neutral EDCs (uncharged) as findings exhibit a correlation between removing neutral EDCs (uncharged) and their molecular width and weight [132,149].

The size exclusion mechanism has been proven to correlate with solutes parameters, such as molecular structure, size, weight, and geometry, including the compound stoke radius [141]. A rise in molar mass was found to be responsible for higher rejection of neutral EDCs in NF. This implies that rejection occurs when neutral compounds with a larger molecular size could not navigate through the membrane pore size, due to the sieve effect of the polymer matrix. Electrostatic interaction between charged membrane surface and organic solutes is also a critical mechanism for the rejection of EDCs contaminants in both RO and NF [134,150], as illustrated in Figure 2.

The electrostatic repulsion between charged organic solutes and charged membrane surface could impede the solutes from reaching the membrane surface, improving permeate quality, and enhance solute removal [142]. The separation of EDCs contaminants via electrostatic interaction can be controlled by either strong electrostatic interactions (ion exchange) or weak electrostatic interactions, namely, hydrogen bonding, dipole-dipole, and ion-dipole interaction through the membrane surface [141,151]. Besides, the estrangement of charged solutes via the electrostatic interface is proportional to the surface charge density for the NF membrane [130]. Simon et al. evaluated removing ibuprofen by NF and RO membranes and observed a direct connection between the electrostatic repulsion, the pollutant, and the pH of the solution [152]. Lowering the pH to values lower than the pK_a_ (acid dissociation constant) of ibuprofen dwindles the electrostatic repulsion, since the membrane turns out to be positively charged, thereby accelerating the attraction of ibuprofen onto the membrane, which poses a negatively charged surface. As reported by Sahar et al. more than 95% of diclofenac was removed by negatively charged RO membrane owing to the electrostatic interaction (repulsion) between the diclofenac and the membrane [153]. In another study, the same interaction was observed when analyzing diclofenac and other compounds, including personal care products with negative charges when in solution, for instance, sulfamethoxazole, ibuprofen, naxoprene, and glimepiride [19]. The removal efficacy weakened noticeably for EDCs containing positive or neutral charges. For instance, the removal was close 100% for naxoprene (negatively charged), against 60% for atenolol (positive) and 20% for acetaminophen (neutral) [154].

The isoelectric point and feed water quality play essential roles in charge exclusion during the separation process [136]. The NF membranes are amphiprotic, determined by their isoelectric point, and most NF membranes are negatively charged. The positive membrane surface charge below the isoelectric point is due to the protonation of amine functional groups, and the negative membrane surface charge is due to the deprotonation of carboxyl groups [155]. The membrane surface charge is also influenced by the humic and ions substances absorbing into the membrane. For example, Ca^2+^ decreases the membrane’s negative charge by creating an outer or inner-sphere complex with its surface [156]. The occurrence of humics improves the membrane’s negative charge, due to its charged nature. Moreover, removing natural organic matter (NOM) could be attributed to electrostatic interaction between the charged humic acid and charged membrane surface at neutral pH. This is confirmed by Rana et al., who observed that steric hindrance appears to be governing the removal of pharmaceutical and personal care products (PPCPs) micropollutants from drinking-water when tighter NF membrane was modified with macromolecules and compared with pristine CA membrane [157]. Authors also reveal that the charged additive seems to imply that charge repulsion also plays a significant role in removing PPCPs.

In recap, the membrane intrinsic properties and frequent combination of removal mechanisms in particular charge effect/electrostatic interaction, size exclusion, and adsorption could facilitate the rejection of EDCs contaminants [123,136]. As elucidated from the above research findings, hydrophobic (pore) adsorption has been reported to play a critical role in UF and MF membrane’s rejection of EDCs contaminants. In contrast, steric hindrance (size exclusion) and charge repulsion are strongly linked with RO and NF membranes, and electrostatic interactions may describe the rejection of polar EDCs contaminants by charged membranes, including UF and MF.

### 2.3. Evaluation of Membrane Processes in the Rejection of EDCs

#### 2.3.1. Evaluation of the Performance of MF and UF Membrane Systems in the Rejection of EDCs

Generally, UF and MF are the most loosened amongst the membrane processes employed in water and wastewater purification in terms of the membrane pore sizes. The UF pore size varies between 0.1 and 0.005 μm and for MF ranges within 5 and 0.1 μm [141]. Previous studies have reported that ultrafiltration (UF) membrane could partially remove EDCs pollutants from different water sources [115,129], as indicated in Table 4. Principally, UF and MF membranes have been regarded as a promising solution to curtail operating costs, since they can be operated at reduced pressure (1–3 bar) and produce more permeate in a shorter period [158]. Nevertheless, it should be noted that microporous membranes cannot eliminate EDCs as efficiently as dense membranes by size exclusion. Thus, consideration should be given to mechanisms, such as adsorption and electrostatic interaction, as the mainstream mechanism strongly influencing the removal of EDCs via UF and MF membranes [141,159]. This is because the adsorption of EDCs in membrane separation is not only constrained to the membrane surface, but may also happen in the porous structure of the membrane and is also explicitly linked to pore radius [125]. Mostly, membranes with bigger pore sizes, such as UF membranes, permit EDCs to approach the inner porous structure of the membrane (with additional sorption sites). Hence, the more porous the membrane, the more EDCs the membrane may permit to adsorb inside the membrane pores together with its surface according to their physicochemical properties.

The UF and MF membrane efficiency in eliminating EDCs could be improved, provided that the adsorption mechanism plays a significant role in retaining EDCs by adsorbing onto the internal pore walls and membrane surface, since adsorption is the principal mechanism for removing micropollutants via UF membranes [117,160]. For instance, Secondes et al. investigated the rejection of EDCs (carbamazepine, amoxicillin, and diclofenac) by a UF membrane using a single hollow fiber membrane component with an effective area of 6.6 cm^2^. The UF membrane with MWCO of 100 kDa displayed low removal (<30%) for all targeted EDCs contaminants [161]. The highest removal was noticed for diclofenac afterward carbamazepine and amoxicillin. This removal trend was associated with their hydrophobic adsorptive properties. Meanwhile, the adsorption of EDCs on the membrane surface is mostly resulting from their hydrophobicity; adsorption was reflected to be the primary rejection mechanism for EDCs removal in this study. Similarly, Pramanik et al. [162] examined the efficacy of a PVDF hollow-fiber UF membrane (pore size 0.1 μm) to remove perfluorooctanoic acid and perfluorooctanesulfonic acid contaminants from lake water. Their findings showed that the UF membrane had minimal removal efficiency for both contaminants (~28 and ~20%, respectively), which was due to the greater pore size of the UF membrane failing to act as a physical impediment for holding the EDCs.

According to the study conducted by Bing-Zhi et al. [163] to examine the rejection of bisphenol A (BPA) using a polysulfone (PS) membrane. The results of their study revealed that the adsorption capacity of BPA towards the membrane is subject to the material, which has an excellent removal rate using polysulfone. The authors evidenced that BPA retention is severely affected by solution pH, thereby leading to a sharp decline in retention when the pH exceeds its pK_a_ value. It was concluded that physical and chemical interactions could energize sorption onto the membrane in hydrophobic adsorption and hydrogen bonding. Some possible limitations observed in this study as polysulfone membranes are highly susceptible to adsorptive fouling, the build-up of many contaminants, and changes in the feed solution matrix that lead to leakages.

Yoon et al. examined the rejection of 27 EDCs and PhACs comprising MWs < 0.4 kDa applying a commercial UF membrane (MWCO −100 kDa). A poor removal (<30%) for all pollutants were noticed, excluding triclosan (>80%), oxybenzone (>70%), erythromycin (>60%), progesterone (>50%), and estrone (>40%) [164]. Their conclusions emphasized that the overall rejection trend was the hydrophobic adsorption of EDCs as a function of K_ow_. Since the adsorption of EDCs onto the membrane surface was dependent on their hydrophobic value, it was believed that EDCs with strong hydrophilic characteristics (less hydrophobic; log K_ow_ < 3) were implausible to be sorbed onto the membrane surface. Nevertheless, EDCs with strong hydrophobicity (log K_ow_ > 3) exhibited the contrary behavior. Numerous other studies also reported analogous trends for removing EDCs using UF membranes [125,165].

Hu et al. investigated the fouling disposition of simulated effluent and the influence of membrane fouling and working pressures on the rejection of selected EDCs during UF membrane filtration tests [166]. The study results showed that the fouled membrane could eliminate (10–76%) of some target compounds. It was observed that cake developed under 50 kPa recorded the least porosity (56.8%), however, it had the superior EDCs rejection efficiency, which could be connected with adsorption and size screening effects. The authors suggested that 50 kPa may be efficient in achieving better EDCs removal with suitable flux.

On the other hand, the rejection of estrogens contaminants from water using MF membranes has been effectively proven, and adsorption capacity ~0.87 µg/cm^2^ using a 0.45 µm polyamide membrane has been reported [57]. However, removing the contaminant from the water was through physical adsorption and a unique chemisorption mechanism.

In recap, both UF and MF membranes are not efficient for rejecting most EDCs in water due primarily to the molecular characteristics of the contaminants, which are often lesser than the pore sizes of MF and UF membranes. Accordingly, the MF and UF arrangements may not function as a hindrance for the EDCs except some preliminary treatment system, including adsorption, coagulation, among others, is accomplished to make the contaminants ‘bigger’ and subsequently permeated. Moreover, MF and UF can be employed in amalgamation with RO as a preliminary treatment to attain more practical removal outcomes and relegate fouling of RO membranes. On the other hand, since adsorption is the principal influential mechanism in UF/MF membranes for the rejection of EDCs, novel techniques to widen sorption sites, enhance surface morphology, and surface manipulation can reinforce the efficacy of UF membranes for EDCs rejection. Membrane surface refinement aimed at diminishing distribution and pore size may as well be beneficial.

#### 2.3.2. Evaluation of the Performance of NF and RO Membranes in the Rejection of EDCs

Previous studies have reported excellent removals of EDCs as high as 99% using membrane technology [141,167,168]. Principally, both NF and RO are elevated-pressure membrane systems with tighter pore sizes lower than 0.01 μm, which needs consistent enormous energy to function. RO membranes are narrower in pore size than NF membranes, and narrow NF membranes are described to be more effectual than loose NF, due to lesser pore size [128]. For example, the non-ionic structure of bisphenol A for the selected pH resulted in its low rejection (74.1%) when compared with ibuprofen (98.1%) and salicylic acid (97%) [169]. Though the molecular weight of bisphenol A (MW = 228 gmol^−1^) is more significant than ibuprofen (MW = 206 gmol^−1^) and salicylic acid (MW = 138 gmol^−1^), the high water partitioning coefficients (pK_a_) value of bisphenol A (pK_a_ = 9.6–10.2) presented a little influence on the electrostatic mechanism by the negatively-charged membrane surface and the lack of electrostatic involvement as in comparison to ibuprofen (pK_a_ = 4.9) and salicylic acid (pK_a_ = 2.9) which possess a negatively-charged/deprotonated form. Hence, unlike salicylic acid and ibuprofen, the principal mechanism for bisphenol A rejection was only the size screening (exclusion). Im J.-K. et al. observed similar results by reporting the size screening effect as the central mechanism for hydrophilic neutral compounds with high pK_a_ [170]. Regarding the positively charged EDCs, the efficacy of rejection is substantially declined owing to their electrostatic interaction with the negatively charged membrane surface and succeeding diffusion.

A comparative study between tight NF to RO by Yangali-Quintanilla suggested that tight NF is a suitable impediment for EDCs, since its rejection performance nearing that of RO and its low maintenance and operational expenses a sustainable project application [171]. It was also stated that the removal rate exceeding 90% are attainable with ‘loose’ NF preceding aquifer recharge and recovery (ARR) in a hybrid system.

Yüksel et al. [150] evaluated the rejection of bisphenol A (BPA) from model solutions using selected RO and NF membranes. Excellent rejection (≥98%) of BPA was achieved with three polyamide RO membranes. Despite these significant removals, high energy demand and too many modular units (membranes) remain the major drawbacks of this study. Hence, the application of this process is not economically feasible, most especially in a full-scale application.

Zielińska et al. [74] combined MF and NF to remove EDCs from biologically treated wastewater. In this study, it was discovered that the two processes achieved complete removal of BPA at an initial proportion of 0.3 ± 0.14 mg/L, and the removal efficiency of 61–75% was recorded for the NF membrane. The authors concluded that the MF membrane appears a favorable panacea for the subsequent treatment of wastewater containing BPA and could be applied at low transmembrane pressure (TMP) than NF. The two significant limitations observed from this study were a decline in the filtration capacity, due to fouling and quick fouling in the MF membrane; thus, reducing the removal efficiency from 37% to 24%. Interestingly, a higher removal efficiency of 97% was reported by Al-Rifai et al. [58] as MF and RO were combined in treating EDCs. However, BPA of a concentration of 500 ng/L was discovered in the effluent after the treatment process, and higher energy demand was required. Similarly, Snyder et al. [172] performed a series of pilot and full-scale dynamic flow-through membrane studies to evaluate endocrine disruptors and pharmaceuticals’ removal rate. Different membrane configurations, including UF, RO, NF, membrane bioreactors (MBR), electrodialysis reversal, were examined in that study. RO filtration was found to be a better technology for the elimination of target pollutants. However, certain compounds could still be present in the RO permeate at trace levels (ng/L). In that study, UF was noticed not to be efficient in removing most compounds, with the exception of some steroid hormones.

Yoon et al. [173] investigated the rejection of EDCs of different physicochemical properties using NF and UF membranes in a filtration process. Results revealed that 30 to 90% of EDCs could be eliminated through the NF membrane compared to only less than 30% observed in the UF membrane. Both steric hindrance and hydrophobic adsorption were the removal mechanisms for removing EDCs in the NF membrane, while the UF membrane entirely relies on the hydrophobic adsorption mechanism for removing hydrophobic EDCs. The authors concluded that the transport phenomenon associated with adsorption is driven by membrane material and water chemistry conditions.

Generally, high-pressure membrane systems, such as reverse osmosis and nanofiltration systems, have been demonstrated to be more suitable for the rejection of EDCs contaminants from water and wastewater, considering their microporous nature and higher flux. For example, Sahar et al. [153] applied RO after CAS-UF and MBR processes to evaluate its removal efficiency in eliminating EDCs. The two processes display the relatively similar and higher rejection of more than 95% for diclofenac and more than 99% removal for most other targeted contaminants. Despite these significant removals achieved in this study, it was observed that concentrations of EDCs ranging between 28 to 223 ng/L were detected in the filtrate from both units. This has proven that RO was not an outright impediment for EDCs. This observation is supported by Steinle-Darling et al. [174], which reveals that NF and RO membranes are still somewhat permeable to some relatively small micropollutants.

The above-reported research findings revealed that the central removal mechanism for RO/NF membranes would be dictated by the interaction between the physicochemical properties of EDCs, membrane material and interfacial characteristics, and the chemistry of the solution. Hence, there is a critical need for more scientific studies to distinguish and elucidate both relative contributions and impact of these diverse interactions and build suitable modeling tools. NF/RO could be a practical approach for removing EDCs contaminants from drinking water and wastewater. Table 4 presents a summary of some research findings on the application of membrane treatment technique in eliminating EDCs pollutants.

### 2.4. Factors Affecting the Membrane Rejection Performance of EDCs

The effectiveness and efficacy of a membrane as a separation barrier are based on several influences, comprising the feed water characteristics, membrane properties, and operational conditions, and physicochemical characteristics of EDCs as illustrated in Figure 4 [120,175,176]. More importantly, there is a need to fully understand the various critical factors that may determine the rejection of EDCs during membrane separation processes.

Physicochemical properties of EDCs are broadly diverse; however, most pharmaceutical compounds, for instance, are polar, biologically energetic, persistent, with relatively strong hydrophilicity to get absorbed in humans, in addition, to avert degradation prior to their curing effect [177]. These properties, together with the small concentration at which they occur and their minute molecular weights or size, make their removal obviously challenging [128].

The physicochemical properties of EDCs in particular hydrophilicity or hydrophobicity, molecular size and weight (length, width, and MW), chemical structure (such as the occurrence of electron-donating or withdrawing functional group), and charge properties are reported to have substantial impacts on their removal by membrane separation [178,179]. It was also reported that larger organics impact and increase the pollutants’ molecular weight in water [180].

Also, the neutral/uncharged EDCs contaminants are typically removed by size hindrance mechanism [181,182], the connection between EDCs contaminants and hydrophobicity rejection assessed as Octanol-water partition coefficient −LogK_ow_) in addition to membrane pure water permeability flux is frequently reported as necessary, exhibiting elevated removal rate for more hydrophobic EDCs, especially in conditions where hydrophobic adsorption removal mechanism is influential [147]. While the impact of water solubility on the rejection is reported to be more insignificant, some other studies indicate that water solubility of a compound should be evaluated as the first investigative parameter on its movement in NF membrane separation as experimentations reveal that neutral (uncharged) compounds with low water solubility, lower molecular weight (Mw), and strong hydrophobicity (high LogK_ow_ values) were rejected better than others with relatively higher water solubility and higher Mw [181].

The physical sieving effect by pores and other physicochemical interactions (including adsorption or electrostatic screening) together with diffusion limitation occur to play a significant role in the rejection of certain EDCs in NF membranes [112]. For neutral (non-charged) hydrophilic EDCs, steric hindrance is most probable the powerful mechanism for rejection. During filtration experiments, the molecular weight/size of EDCs contaminants connected well with the mean membrane pore distribution [183,184]. The negatively-charged hydrophilic EDCs can be further removed by electrostatic repulsion via negatively charged membrane surfaces [97,126,185,186].

Hydrophobicity has been reported to have a significant contribution to the adsorptive rejection mechanism of the membrane. This is because the adsorption mechanism has proven to be linked with solute-hydrophobic interactions [135]. Their contact angle consistently characterizes membrane hydrophobicity, though, hydrophobicity of solutes can be quantified and correlated with the logarithm of the octanol-water partition (LogK_ow_) [143]. Typically, hydrophobic compounds are compounds with LogK_ow_ ˃ 2, whereas octanol/water partition coefficient values are expressed as log [ratio of the proportion in the octanol state to the proportion in the aqueous state at varied pH], such that the dominant state of the compound not ionized [187]. Hydrophobic adsorption at the initial operational period is controlled by the hydrophobicity of compounds to be treated [135]. Hence, hydrophobic properties could influence the adsorption mechanism where potent hydrophobic compounds, particularly alkyl phthalates, non-phenolic pesticides, and aromatic pesticides, were strongly eliminated via minimal desalting membrane [126]. The membrane’s pore size could undermine estrogenic compounds’ rejection, whereas rejection by porous membrane declines with sorption.

The membrane characteristics also play a dominant role in accelerating the removal of EDCs contaminants. These characteristics may include membrane’s permeability measured as flux, pore size/MWCO, surface morphology measured as roughness, surface charge determined as zeta potential, and hydrophilicity or hydrophobicity—measured as water contact angle [120,132,188].

Porosity has been considered another functional parameter in several studies to assess EDCs rejection in membrane separation [183,189,190]. Košutić et al. [28] investigated the porosity of some commercial NF and RO polyamide TFC membranes. They stated that the membrane’s porous structure was an influential variable in defining the membrane performance and that EDCs rejection could be described by membrane porosity parameters (such as N and PSD). The actual number of pores, N; in the skin layer of NF and RO membranes rise with an upsurge in pressure, and PSD could be modified under elevated pressure, since some membranes were more responsive to pressure variations than others and displayed different removal performance.

The membrane fluxes and transmembrane pressures vary, and largely depend on the working pressures of various membrane processes (such as UF, MF, NF, and RO). The lower-pressure membranes, namely, UF and MF, require lower operational pressure with higher fluxes and lower TMP, whereas, high-pressure membranes (NF and RO) need higher pressures and present higher fluxes and higher TMP. Thus, the former requires lesser energy than the latter [153]. The operational pressures in NF are comparatively lower than the RO, thereby filtration takes place at a minimal energy requirement (21% less than RO), and superior water fluxes can be achieved at reduced transmembrane pressures (TMP) [191].

The feed solution conditions, for example, pH, the concentration of natural organic matter, and ionic strength, are outlined to influence the efficacy of the EDCs rejection [145]. Ionic removal is attributable primarily to the electrostatic repulsion between the membrane’s surface charge and the charged EDCs contaminants. In addition, the NF process is more responsive to pH and ionic strength of feed water than RO [191].

The nature and characteristics of feed water play a pivotal role in eliminating EDCs pollutants from water during the membrane process. The copresence of natural organic substances in water could moderately undermine the efficiency of EDCs removal, due to the competition for the insufficient adsorption sites with EDCs. Thus, resulting in a minor decrease in EDCs removal [117,168]. As illustrated in Figure 2, the adsorption of EDCs onto the membrane largely depends on the pH, and the deprotonation of carboxylic and or sulfonic acid groups usually contributes to the negative charge on the membrane surface [120,192].

Solution chemistry, particularly pH, influences the mechanism of EDCs removal and general rejection, since specific properties (for example, acid dissociation constant pK_a_) of EDCs in the feed water could transform extensively with adjustment in pH. Nghiem et al. [145], while utilizing ‘loose’ NF to remove carbamazepine, ibuprofen and sulfamethoxazole, observed low and inconsistent EDCs removal efficiency and inferred that removal of EDCs was strongly affected by an adjustment in solution pH, which alters the ionic strength and charges of targeted EDCs. This trend is also observed in other studies [193,194]. Besides, the occurrence of humic acid in the feed matrix, for instance, can boost the rejection of EDCs as indicated by De Munari et al., where removal of pesticides endosulfan with Mw less than MWCO of NF membranes enlarged with the presence of humics in feed matrix [195]. In fouled NF membranes, steric hindrance (size screening) was noticed to be the primary removal mechanism, and the potential of improved adsorption of EDCs to fouling material could arise [171,188].

Different pH conditions will considerably modify the membrane surface charge. In this context, it should be noted that increasing pH will upsurge the negative surface charge of the membranes, thereby enhancing significant rejections, particularly for negatively charged compounds [196]. Studies from [197,198] also revealed that the influence of the pH on the membrane surface charge is attributed to the separation of functional groups and the charge of the target compounds, which can be modified by altering the solution pH exceeding pK_a_ value to increase the rejection. Besides, several studies have reported that increasing the pH and the deprotonation of functional groups are responsible for the increasingly negative charge in zeta potentials for most membranes [198,199]. Thus, deprotonation of EDCs causes charge repulsion between EDCs contaminants and negatively charged surface of the membrane, thereby interfering with the sorption of EDCs onto the membrane surface and facilitate a decline in the rate of removal [163]. Changes in pH not just modify the surface of the membrane yet influence the state of electrolyte solutes separation, solubility, and orientation of the solute [198]. It can be deduced that pH is an indispensable factor dictating the removal rate. A low or high pH triggers the membrane surface to be more charged, enabling polar compounds to simply undertake dipole-dipole interactions, thereby increasing removal [200].

Besides, a low or high pH reduces the membrane pore size, since the charged groups create a stretched chain conformation [156], and a low pH initiates the acidic hydrolysis of chemical bonds and decreases the degree of crosslinking, which has a negative influence on contaminants removal [201].

### 2.5. Membrane Fouling Challenges

Generally, membrane technology performance could be plausibly undermined from two consequences: Membrane fouling and concentration polarization. Concentration polarization refers to a surge in the concentration of the unwanted dissolved or suspended solids near the membrane surface, as illustrated in Figure 2. In contrast, the membrane fouling phenomenon refers to the obstruction of membrane pores by several adsorption and sieving of compounds and particulates within the membrane pores or onto the membrane surface during filtration. Pore blockage lowers the production volume of permeate and raises the membrane filtration process [202,203]. Membrane fouling is attributed to the effect of irreversible accretion of dissolved or suspended solids on the exterior membrane surface or membrane pores, thereby jeopardizing the overall efficacy of the membrane [204]. However, the principal challenge and the most critical drawback is the membrane fouling, which is the leading constraining factor in commercial membrane practical applications [205]. However, the membrane is prone to surface fouling during operation, which significantly hampers membrane performance in permeability and selectivity [96,206].

Notably, both permeate flux and transmembrane pressure (TMP) are the key indices of membrane fouling. Since fouling results in a substantial rise in hydraulic resistance, which could be demonstrated as a reduction in permeate flux or TMP rise when the system is operated under consistent-flux or steady-TMP situations [206,207]. The accretion and adsorption of organics within the membrane pores and even on the membrane’s surface during the filtration process raise the TMP (transmembrane pressure) [208,209]. In a process where the permeability is controlled by raising TMP, the energy desired for filtration rises. During a prolonged operation period, fouling is not entirely reversible as a result of backwashing. Even as the number of filtration cycles continues to increase, the irreversible proportion of the membrane fouling is also rising. To attain the required output volume, the membrane needs to be chemically cleaned to restore much of its permeability [203].

Generally, fouling can be categorized as reversible or irreversible (permanent) and backwashable or non-backwashable, depending on the particles’ strength attached to the membrane surface. Physically reversible fouling occurs by accretion of sludge particles whose particle sizes are bigger than the membrane pore size can be annulled via physical cleaning (such as surface cleaning, backwash) [210]. Irreversible (permanent) fouling g is considered to develop via the attachment of solutes and colloids within the membrane pores and cannot be eliminated by chemical cleaning, backwashing, flushing, or any other procedure, and the membrane cannot be reverted to its initial condition [203,210]. On the other hand, backwashable fouling can be eliminated by altering the path of permeate flow via the membrane’s pores at the end of each filtration cycle. Non-backwashable fouling cannot be prevented by regular hydraulic backwashing throughout filtration cycles [203]. Nevertheless, the non-backwashing of the membrane can be conducted by chemical cleaning.

Fouling can also be classified into four categories, namely: Scaling fouling/inorganic, colloid/particle fouling, biological/microbial fouling, organic fouling, based on the nature of fouling material. Scaling or inorganic fouling is triggered by particles’ agglomeration when the concentration of the chemical species surpasses its saturation concentration. Several studies have shown that a higher concentration of magnesium (Mg^2+^) and calcium (Ca^2+^) provoked more fouling [211,212]. Conversely, organic fouling could be stimulated by the blockage of the membrane by organic carbons, and organic substances usually assemble on the inner surface of the membrane [203]. Based on the evaluation of the extracted solution during chemical cleaning, it was discovered that most soluble organic foulants had low molecular weights, and calcium was the dominant inorganic foulant [213].

Similarly, quite a few research findings have indicated that natural organic matter (NOM) is the main foulant in the ultrafiltration membrane and that the different constituents of NOM aggravate different types of fouling [214]. According to Makdissy et al., the organic colloid portion produces severe fouling [10]. However, polysaccharides are regarded as the leading foulant [215]. Other studies also indicated that most of the fouling is elicited by hydrophobic NOM components [216]. However, neutral hydrophilic NOM constituents have been reported as significant foulants by some researchers [217].

### 2.6. Mechanism of Fouling in Membrane Filtration

Figure 5 shows the relationship between permeability flux and time as regards the fouling mechanism. As illustrated in Figure 5, a typical flux (permeability)-time curve of ultrafiltration (UF) begins with (I) a fast-initial decline of the permeate flux, (II) succeeded by a prolonged period of slight flux decrease, and (III) completed with a stable-state flux. The decline in the membrane flux during filtration is due to the upsurge in the membrane resistance via the growth of a cake layer on the membrane surface and membrane pore occlusion. The blockage of the membrane pore enhances the membrane resistance, while the cake formation forms a supplementary layer of resistance to the permeate flux. Hence, cake formation and pore blocking can be regarded as the two critical mechanisms for membrane fouling [203].

The rapid initial decline in permeate flux could be due to the rapid blocking of membrane pores. Maximum permeate flux often occurs at the start of the filtration period, since the membrane’s pores are open and clean at that time. Flux reduces as the pores of the membrane are filled by the accumulated particles. Pores are more prone to be partially obstructed, and the magnitude of pore blockage relies on the nature and comparative size of the particles and pores. Clogging is generally more complete when the particles and pores’ shape and size are identical [219,220]. The blocking of pores is a fast process compared to the development of cakes, because less than one layer of particles is required to attain complete blocking [218]. Additional flux decline after pore blockage results from the formation and growth of a cake layer on the membrane surface. The cake layer is formed on the membrane surface as the amount of retained particles increases. The cake layer creates additional resistance to the permeate flow, and the resistance of the cake layer increases with the growth of cake layer thickness. Therefore, the permeate flux remains diminishing with time [203].

In an MBR system, fouling anomaly results from an interaction between sludge suspension constituents and the membrane material. Though the aerobic MBR membrane can typically be used in the AnMBR process, since the sludge suspension in the AnMBR system is considerably different from that in the aerobic chamber, thus, presenting individual-specific influences on membrane fouling characteristics [221]. Fouling could result in regular cleaning, which could shorten the longevity of the membrane, leading to higher operating and maintenance expenses and increase energy demand for sludge recirculation or gas scouring [105,222]. The accretion of dissolved or suspended substances on the membrane exterior surfaces, its pore openings, or within its pores could undermine the efficacy of a membrane [223]. These foulants could be solutes, colloids, and suspended particulates (cell debris and microorganisms) in the MLSS [224,225,226]. The Physico-chemical interactions that occur between the membrane material and the foulants result in membrane fouling. Inability to effectively regulate the fouling of the membrane in MBRs could perhaps, in certain situations, result in failure to treat the essential design flow [227].

Fouling in MBRs takes place in various ways, such as cake formation, pore-clogging, and pore narrowing. Pore clogging occurs due to the blockage of the micropores of the membrane by the foulants. Pore clogging considerably relies on the membrane pore size and size of the solute particle. The adhesion of the materials to the pores is facilitated by sticky substances present in the solution. On the other hand, cake formation emanates from the continuous build-up of inorganic matter, biopolymers, and bacteria clusters, which create a biocake layer on the surface of the membrane [228]. The cake layer boosts membrane filtration resistance. The schematic representation of membrane fouling mechanisms in MBRs is demonstrated in Figure 6.

Operationally, membrane fouling diminishes the permeability flux when the MBR is run at steady transmembrane pressure (TMP) and leads to a surge in TMP when the MBR functions at stable permeate flux. Moreover, at a stable flux process, a sharp rise in TMP implies critical membrane fouling. This rapid TMP surge is referred to as “TMP jump.” TMP jump has been illustrated as a three-stage process [229,230]: Stage 1—an initial “conditioning” fouling, which is triggered by initial pore blocking and solutes sorption; stage 2—linear or feebly exponential steady rise in TMP owing to an accumulation of biofilm and additional membrane pore blocking; and stage 3—a sudden surge in the rate of TMP rise (dTMP/dt) [3]. Stage 3 shall be considered as the consequence of severe membrane fouling, and thought to be due to sequential closure of pores and fluctuations to the local flux caused by fouling, which triggers local fluxes to surpass the critical value, thus, accelerate the deposition of solute particles [231,232] and rapid changes of the cake layer structure [229]. Bacteria within biofilms appear to perish thanks to oxygen deficiencies, and hence, release more EPS [233]. Once stage 3 occurs, membrane cleaning is necessary. The practical inference of this is that a delay in stage 3 will permit for a decline in membrane cleaning rate, which will ultimately lead to a reduction in MBR operational expenses. Hence, one significant purpose of fouling control is to hamper TMP jump via reduced working flux or alteration of sludge properties (EPS content, MLSS, apparent viscosity, and floc size) [234].

Currently, fouling extenuation receives significant research interest to mitigate these challenges by modifying existing membranes to enhance their anti-fouling capacities and develop novel membrane constituents with superior fouling tolerance [235,236]. The ultimate objective of membrane fouling studies is to establish techniques for containing membrane fouling and membranes’ maintenance. Fouling mitigation techniques based on the membrane fouling parameters include preliminary (feed) treatment, membrane transformation or modification and optimal membrane design, purification, maximization of working conditions, air stripping, enhancement of hydrodynamics of the separation processes, and other fouling-resistant application techniques [206,237]. To minimize the possible membrane fouling, wastewater sewages could be subjected to several preliminary treatment procedures before UF, NF, or RO membrane processes, but not rarely applicable to drinking water [112]. The main target of the preliminary treatment is to upgrade the quality of the crude feed water in the case of wastewater and also extend membrane lifecycle by declining fouling, precipitation, and scaling [206]. In addition, the design of UF and MF as a preliminary treatment procedure to RO and NF is a conventional system with the benefit of relegating fouling manifestation in RO/NF and upgrading the performance of the treatment systems. This is appropriate to water reclaim setups and desalination procedures [128].

The optimization of the membrane operational conditions within the design limits of the various membranes is critical for reducing fouling. The conditions, such as hydraulic pressure, temperature, pH, and hydrodynamic condition, can be engineered to lower the fouling evolution [206].

Air sparging is one technique used for fouling mitigation, particularly during the UF process [165]. Typically, the air is applied during backwash to facilitate the removal of foulants from the membrane surface during drinking water treatment. Previous research has established that air sparging during membrane separation may also be a low-cost method of relegating fouling in wastewater [238], as well as natural water [239], and synthetic water matrices [240]. Furthermore, air sparging may be enhanced (bubble frequency and size) to enable fouling mitigation. Several studies have pinpointed big (>100 mL) ‘pulse’ bubbles as an efficient air sparging procedure for UF fouling reduction [239,240,241,242]. Moreover, pulse bubble sparging is an energy-effective in contrast to traditional coarse bubble sparging [241]. Air sparging during membrane separation relegates fouling by stimulating shear stress at the membrane surface, which turns to back-passage foulants (biopolymers). According to the interactions between EDCs and natural organic matter (NOM), both in solution and within the fouling layer, the surface shear stress provoked by air sparging may influence contaminant retention during UF of natural waters. The increasing knowledge of turbulence variations has yielded innovative aeration patterns for the air sparger configuration, which tends to optimize the capacity of altering hydrodynamics to reduce fouling [237].

Furthermore, a new research focus has emerged based on membrane modification. Such a technology development may modify membrane properties with nanoparticles to fine-tune efficiency for types of EDCs contaminants and equally advance fouling resistance.

Modification of membrane is another possible, attractive, and efficient panacea to overcome membrane fouling drawback, resolutely hampers advancement in membrane filtration technique [243,244]. Extensive efforts have been dedicated to mitigating membrane fouling from different studies. Recently, the development of improved membranes to enhance the removal of recalcitrant EDCs pollutants from water has received considerable attention, since membranes are usually reported to have the capacity of eliminating nearly 90% EDCs [159]. Conversely, it is imperative to note that membrane surface modification could enhance membrane surface properties while maintaining the membrane backbone unmodified, optimizing, and accelerating EDCs removal efficiency.

Surface modification for hydrophilic membrane enhancement is the most widely accepted technique for mitigating membrane fouling, since their surface properties are the most influential characteristics of the membrane materials [176,245,246]. This modification could be undertaken via surface mixing (blending), surface grafting, surface coating, and plasma treatment to incorporate polar organic functional groups on the membrane’s surface. Following CO_2_ and NH_3_ plasma treatments, membrane hydrophilicity was found to upturn significantly, and modified membranes exhibited enhanced filtration performance and flux recovery than those of unmodified membranes [247].

An extensive review by [245,248,249] emphasized that various techniques, including physical modification, such as surface coating blending, sol-gel technique, and surface grafting (chemical modification), could transform the membrane surface properties. On the other hand, membrane surface could be enhanced through a sol-gel approach [250,251], in-situ growth of nanoparticles, or in situ polymerization in a polymer matrix [243,252]. In addition, Kango et al. reported that the membrane modification approach could strengthen the interfacial contacts between polymer matrices and inorganic nanoparticles, thereby providing unique characteristics in the membrane, including enhanced anti-fouling resistance, together with filtration efficacy [246].

Blending involves the direct incorporation of the nano-scale particles into the membrane polymer [203,253]. Inorganic nano-scale particles (NPs) generally exhibit superior surface energy [254], making them more hydrophilic, therefore, highly suitable for membrane modification via blending. This technique is facile, displaying beneficial viability in practical utilization, and is frequently utilized to produce inorganic/polymer nanocomposites. Usually, the mixing can be prepared via solution blending or melt blending [253,255].

However, attaining an efficient dispersal of the nanoparticles in the polymer matrix, due to high agglomeration tendency remains a big challenge for the blending process [246]. This problem could be addressed by the careful introduction of a suitable quantity of nanoparticles into the mixture and modification of the nanoparticles with sodium dodecyl sulfate (SDS) [255]. For instance, Muhammad et al. reported that the utilization of SDS solution to transform SiO_2_ nanoparticles achieved a higher reduction in the nanoparticle’s agglomeration, since the presence of polar group is likely to yield steric repulsive forces, lowering the contact between nanoparticles and enhancing nanoparticles dispersion property. A similar observation was also recorded from other studies [123].

Modi and Bellare [256] reported higher mechanical strength, hydrophilicity, excellent surface charge, and superior anti-fouling when copper sulfide/carboxylated graphene oxide nanohybrid was blended with neat PES membrane polymer. Greater flux recovery (90.1%) was also recorded in this study.

The surface coating can be carried out by coating the surface of the membrane with a thin film layer of additives, as shown in Figure 7a [248]. This method can be performed via coating with a monolayer utilizing similar techniques or Langmuir-Blodgett, physical adsorption by non-covalent bonds [257], and simultaneous spinning, such as triple orifice spinneret, among others. The physical adsorption technique comprises the attachment of a thin hydrophilic layer on the surface of the membrane to enhance the membrane [258]. Adsorption occurs via hydrophobic interactions, electrostatic interactions, chemical and hydrogen interactions with functional groups on the membrane [258]. For instance, Hou and his co-workers [259] performed surface coating of PVDF membrane with titanium dioxide (TiO_2_) nano-scale particles at a lower hydrothermal temperature sol-gel approach, preceding immobilization of laccase on the membrane surface via a chemical coupling. Because of this, substantial improvement of BPA removal with lower fouling tendency was reported for the modified membrane. Cheng et al. [260] study the synthesis of coated PES membrane with polydopamine (DPA) in an alkaline solution of dopamine. The outcome of the study revealed that surface coated membranes have a superior anti-fouling effect than the unmodified membrane. The coating procedure can be easily applied, and it is a means of efficiently enhancing the membrane without damage to the mechanical property of the membranes.

Nevertheless, the significant shortcomings of this technique are that the coating of hydrophilic monomers and polymers can be deteriorated along with prolonged usage of the membrane [261]. Besides, this approach is environmentally harmful because it requires applying chemicals under hazardous and unsafe conditions to attach the hydrophilic polymer on the membrane surface [262]. This approach may further contaminate the treated water. Figure 7a demonstrated the surface coating of the upper membrane layer with nanoscopic polydopamine (PD) that can efficiently improve the hydrophilicity of the membrane.

The sol-gel technique involves introducing nanoparticles into the polymeric solution that develops interpenetrating polymer networks between the polymers and the nanoparticles under moderate conditions [159]. The technique is undertaken by dissolving nanoparticles into a polymeric solution, and consequently, gel formation [237,263].

The strong interfacial interaction and strong compatibility developed between inorganic and organic phases can enhance the hydrophilicity, mechanical, thermal, and chemical stability of the membrane, thereby facilitating the improved performance of the modified membrane [246,264].

Surface grafting is carried out by grafting synthetic polymeric materials onto the surface of the substrate to increase the surface geometry and chemical functionality of the intrinsic organic and inorganic materials, as demonstrated in Figure 7b [246]. The polymeric chains can be grafted to the surface of nano-scale particles either through an in situ monomer polymerization (grafting-from) technique where polymeric chains are grown from immobilized initiators or the attachment of covalent end-functionalized polymers to the surface (grafting to) as illustrated in Figure 6b [265].

Zhu et al. [266] tailored adsorption behaviors and mechanical properties of a composite membrane toward BPA via ultraviolet (UV) irradiation graft polymerization. The study showed that the membrane could achieve 80% BPA removal and other EDCs with different characteristics through adsorption onto the grafted surface. They also found that the composite membrane possesses a better double-layer structure without delamination, and the used membrane can be quickly recovered and reused. Baransi-Karkaby et al. [243] modified commercial low-pressure RO membrane by grafting poly(glycidyl methacrylate) to enhance the rejection of multiple micropollutants from water. Hou et al. [267] reported significant improvement in the membrane hydrophilicity and the pure water flux compared to the pristine membrane when anti-fouling and antimicrobial characteristics of the UF membrane were examined using an engineering thermoplastic during surface grafting procedure. They also concluded that PEO and Nchloramine modified membranes are capable of resisting membrane fouling.

### 2.7. Nanomaterials and Membrane Technology

There is no doubt that polymeric membranes are highly susceptible to fouling, which is a major drawback for the efficient operation of membrane systems in recent years. Considerable reviews have been dedicated to advancing the utilization of nanoparticles in membrane technologies [246,250,253,268,269].

This enormous interest by researchers has triggered several studies utilizing nanomaterials, such as zeolite-zinc oxide, poly-4vinylpyride b-ethylene oxide, mono-porphyrin, polyamide nanotubes, carbon nanotube (CNT), polypropylene, MnO_2_, nanofibrous, TiO_2_, PA/TiO_2_, and AgBiO_3_, via incorporation in the membrane matrix, as seen in Table 3, to improve the properties of the membranes and increase the rejection rate of EDCs contaminants [21,270,271,272,273,274,275]. Findings from their studies have reported significant improvement in the membrane properties, including excellent hydrophilicity, higher flux, improved membrane strength, wettability, stability, enhanced mechanical properties, better reusability, and upturn rejection of EDCs contaminants (˃99%). Notably, significant removal (˃98%) of BPA and other compounds was recorded when nanomaterials were incorporated in the membrane matrix in various studies [275,276,277,278], as indicated in Table 4. In the same vein, Alvarez et al. [279], in their review, reported that some distinctive properties of nanomaterials (such as high hydrophilicity, reactivity, and small size) could be explored to mitigate membrane fouling.

The pursuit of novel nanomaterials inclined to carry out multiple processes concurrently is another research area that could be further explored. For instance, in the work of Fischer et al. [263] where titanium dioxide (TiO_2_) (semiconductor nanoparticles) were synthesized and applied on the surface of hydrophilic membranes. In their study, incorporating non-aggregated, highly bonded TiO_2_ nanoparticles onto the membrane surface was found to enhance its anti-fouling properties and demonstrated an intensely active ability to oxidize pharmaceuticals compounds photo-catalytically.

Interestingly, titanium dioxide (TiO_2_) nanomaterial is not the only available semiconductor capable of producing materials with novel properties. Hence, other semiconductors, such as zinc, copper, silver, or platinum-based materials, should also be researched to improve the capacity of the composite membrane to reject EDCs contaminants during membrane processes.

From the foregoing, various studies have established that all the membrane processes could eliminate EDCs from water. Higher rejection of EDCs could be achieved via the application of high-pressure driven membranes, particularly RO, NF, and FO, considering size exclusion mechanism. However, high energy demand and associated costs in RO and NF have reduced the broader application of these systems.

Notably, a membrane with larger pores, particularly UF and MF, can also be considered if the main removal mechanisms associated with the process are adsorption and electrostatic repulsion.

Hence, the application of UF and MF membranes with relatively lower energy demand, due to its lower pressure intimately connected with low-cost, deserves more attention to the treatment and elimination of EDCs. This is because the UF technique has a viable market demand in advanced water/wastewater treatment. Their performance could be strongly enhanced via modification of the membrane surface to significantly eliminate the EDCs from water and mitigation of fouling, without compromising the flux and permeability. Comparatively, blending is the most effective and conventional technique of modifying membranes by directly mixing inorganic additives into the polymeric matrix and other modification techniques. Though adequate dispersal and homogeneity of nanoparticles in the polymer matrix are somehow challenging to achieve, surface modification of nanomaterials and addition of sodium dodecyl sulfate (SDS) solution prior to blending is strongly recommended to minimize the conglomeration of nanoparticle and enhance its dispersibility of the dope mixture [123,246,256].

### 2.8. Removal of EDCs by Membrane Bioreactor Processes

In the last few years, membrane bioreactors (MBRs) treatment technology has received global attention as a promising method for treating various wastewater and has emerged as a better choice over conventional municipal wastewater treatment processes [280].

The membrane bioreactor is a hybrid technique that integrates the physical separation of a membrane with the biodegradation of a conventional activated sludge process at the same time. Low-pressure membranes with a higher molecular weight cut-off are frequently employed in the system owing to reduced energy demands consumption [160,281]. Comparatively, the MBR system has been proven to exhibit several distinct advantages—such as operational robustness, higher microbes population in the immediate surroundings of the membrane surface, guaranteeing the extermination of the EDCs contaminants before the membrane filtration process, simple automation (if properly designed and maintained), lower volume of sludge generation, minimal footprint capacity of the treatment unit, and complete containment of solids—compared to conventional treatment systems [137,282,283,284]. Besides, owing to the steric hindrance of the membrane, contaminants with a molecular weight higher than the molecular weight cut off (MWCO) of the membranes are held, thus, ensuring better contact with the degrading microbes within the MBR for its complete elimination [282].

Notably, improved MBR methods are becoming increasingly desirable to address weaknesses, such as poor EDCs removal rates, energy demand, and membrane fouling [285]. Previous research has established that MBR can remove contaminants with complex structures, such as EDCs, pesticides, and pharmaceutical substances [286,287]. The recent advances in wastewater treatment technologies have witnessed the incorporation of membrane technology into biological treatment processes to enhance the removal of various EDCs pollutants from different water sources (Table 5).

According to Radjenović [148], MBRs effectively remove a wide range of micropollutants, such as EDCs, especially compounds that are not susceptible to activated sludge systems. This can be attributed to the fact that prolonged SRT in MBRs could enhance better microbial degradation of the contaminants, conserve sludge in which several compounds adhere, and the membrane surface’s ability to impede the compounds.

More importantly, the need to attain higher removal of EDCs contaminants requires an in-depth understanding of the removal mechanisms and factors that control the removal of EDCs contaminants from wastewater during the MBR process. These factors include sludge concentration and age, wastewater chemistry, pH, operating temperature, the existence of anoxic and anaerobic chambers, and conductivity [288].

#### 2.8.1. Mechanisms of EDCs Removal during MBR Process

To achieve a higher EDCs removal rate during the membrane filtration process, it is crucial to understand the removal mechanisms and factors controlling the removal of EDCs from wastewater using MBR technology to ensure better process efficiency. In this context, removing EDCs contaminants from water and wastewater using the MBR system is based on the physical barrier by the membrane, sorption, biodegradation, photo-transformation, and air stripping [289,290].

Notably, the fate of EDCs contaminants during MBR treatment is strongly governed by both biodegradation and sorption [119,291]. Microbial decomposition (biodegradation) and adsorption mechanisms can be compromised by several elements and can occur synergistically or separately [281].

Biodegradation involves microbial (such as algae, bacteria, fungi) degradation of organic compounds into elementary chemical structures, often leading to complete mineralization [292]. It is the principal mechanism for removing polar EDCs contaminants, with extremely low sorption, and could be enhanced under prolonged SRT conditions [293]. In contrast, sorption succeeded by membrane retention of the solids is the dominant mechanism responsible for removing apolar contaminants [294]. Sorption signifies how EDCs contaminants turn out to be connected with the solid phase. It has been stated that the sorption of EDCs contaminants against sludge flocs (and certain microbial products) is one of the crucial factors dictating the removal of EDCs contaminants in the wastewater treatment process [148], while the performance mainly depends on the sorbents (such as mass concentration) and the physicochemical properties of EDCs contaminants (such as hydrophobicity, electrostatic interactions, and hydrogen bond) [295,296]. The lipophilicity (logD_pH_) of EDCs contaminants can considerably affect their sorption propensity to activated sludge. The more hydrophobic the compound is, the more significant the sorption propensity, and the higher the removal efficiency is generally anticipated [297]. Several studies have revealed that MBRs display superior removal efficiency for the comparatively hydrophobic EDCs (logD > 2–3) than the hydrophilic (logD < 2–3) [179,298,299]. Intermolecular forces can also influence sorption in the hydrophobic adsorption condition, EDCs contaminants transport from the aqueous phase (i) into the lipophilic cell membrane of microbes in activated sludge and (ii) hydrophobic surfaces of the sludge medium. This process can be defined by the values of K_ow_ [300], while pK_a_ is also an essential factor because if the chemical is mostly ionized, the hydrophobic interaction will be diminished.

Sorption towards the membrane has very little significance considering the smaller molecular size of EDCs contaminants than the membrane pore size, which presents an insufficient sorption site and inadequate membrane surface area available for sorption. However, further elimination of EDCs contaminants through the MF membrane of the MBR could be attained as a result of the production of the ancillary secondary layer by the deposition of EDCs contaminants (secondary barrier) on the membrane [153,301]. The removal of hydrophobic contaminants is primarily controlled by the bio-sorption of the micropollutants to the activated sludge, while hydrophilic contaminants removal is predominantly determined by bio-sorption followed by biodegradation [179,291]. Hence, when the system SRT is sufficiently increased (not less than 8 d), the removal of EDCs via sorption and later biodegradation can be improved [299]. Compounds containing toxic groups and complex structure (including nitro groups and halogens), still can display strong resistance to biodegradation and tend to have very minimal removal [215]. Besides, the presence of several degraded metabolites/intermediate by-products infringes its injurious effects on the removal mechanisms by sorption and biodegradation [282].

Several operational factors are responsible for the performance of the MBR system for the elimination of EDCs from wastewater. These factors involve biological factors (such as concentration and age of the sludge, wastewater constituents, and other physical parameters, such as temperature, pH, conductivity, anaerobic and aerobic, or anoxic environment predominant in the membrane compartments), wastewater characteristics (in particular ionic strength, pH, and organic matter concentration), operating conditions (pH, SRT, HRT (hydraulic retention time), temperature, redox condition), membrane characteristics (hydrophobicity, porosity, flux), and physicochemical properties of the contaminants (chemical/molecular structure, molecular weight).

Some of these factors can undermine the operational efficiency of wastewater treatment using MBR and influence the biodegradation and adsorption mechanisms responsible for removing EDCs in MBRs [281,282,302]. This instead causes an inconsistent and undefined removal efficiency. Studies have shown substantial variation in removing EDCs pollutants by MBRs, ranging from almost complete elimination for some compounds (namely bezafibrate, salicylic acid, and ibuprofen) to almost zero removal for several others (such as diclofenac and carbamazepine) [215,286]. For instance, the removal efficiency recorded for 15 endocrine-disrupting compounds, each at a concentration of 1–5 μg L^−1^ by MBR technique, varied from 92% to 99% [303]. In an MBR study conducted by Trinh et al. [304], diverse removals of PPCP compounds ranging from −34% to >99% with 23 PPCPs recorded ˃90% removal. The authors revealed that very concentrated compounds in the influent were significantly eliminated. The authors suggest that the removal rate of the target compounds could be improved under various working conditions, including longer SRT and HRT. Similar findings from Jiang et al. also reported fluctuated removals of 22 selected EDCs pollutants (11.0–99.5%) during the MBBR-MBR operation [305]. Personal care products (PCPs), such as salicylic acid and propylparaben, were removed entirely using an MBR system. Removal efficiencies of 99%, 97%, and 70–80% were obtained for triclosan, atenolol, and beta-blockers, respectively, using this system [306].

#### 2.8.2. Performance of MBRs in the Removal of EDCs from Water

Guerra et al. [306] examined the effects of working conditions on eliminating BPA in large-scale wastewater treatment facilities (WWTFs). The study results showed that MBR processes displayed good performance with removal efficiencies ranged between 1 to 77%. Authors found that biodegradation and sorption were accountable for removing BPA, and both could be affected by working conditions, namely, mixed liquor suspended solids (MLSS), solids retention time (SRT), and hydraulic retention time (HRT). Moreover, they suggest achieving over 80% removal efficiency required conditions of (HRT: 13 h, SRT: 7 days, MLSS:1600 mg/L) during summer and (HRT:13 h, SRT:17 days, MLSS:5300 mg/L) during winter must be fulfilled, respectively. Though the chemically assisted primary treatment achieved very low BPA removal and the seasonal variation could undermine the performance of this process.

Significant removals of 93.9% and 98% of BPA were achieved by both conventional MBR and forward osmotic MBR in a study conducted by Zhu and Li [307] to investigate the elimination of bisphenol A (BPA) from synthetic municipal sewage using a combination of a conventional membrane bioreactor and forward osmotic MBR sharing a single reactor. The problem of salt leakage and lower retention of MF (10%) has restricted the potential application of this study. In another related study, Kim et al. [308] reported complete removal of naproxen, acetaminophen, and caffeine through biodegradation in a Canadian full-scale MBR system. Though the authors concluded in their study that removal efficiencies of pharmaceutical EDCs in MBR were diverse, fluctuating from −34 to >99% among the 99 pharmaceutical compounds investigated.

Trinh and co-workers [304] revealed that diclofenac, trimethoprim carbamazepine have been recognized as stubborn compounds that are not easily removed via MBRs with diverse removal efficiencies in the literature ranging from 0 to 50%. This anomaly is because they are difficult to biodegrade and poorly adsorb to biomass [148,286].

Li et al. [302] investigated instantaneously activated carbon adsorption in an MBR to enhance the removal of carbamazepine and sulfamethoxazole. Their findings reveal that the addition of 1.0 g/L (PAC) considerably increases the removal rate of the target compounds to 92% and 82% from 64% and negligible removal in the MBR system, respectively. The higher removal recorded was attributed to the higher adsorption affinity of the compounds towards the PAC.

Notably, a predictive model was developed by Wijekoon et al. [309] to evaluate the elimination of trace organic EDCs during wastewater treatment using AnMBR. The findings of the research demonstrated a strong correlation between hydrophobicity and molecular characteristics of EDCs (such as electron-sharing groups (ESGs) and electron-removal groups (ERGs), particularly those having Nitrogen and Sulphur. Besides, all hydrophobic EDCs pollutants were adequately removed (˃70) using AnMBR regardless of their molecular characteristics, while hydrophilic EDCs with electron-donating functional groups were sufficiently removed (˃70). However, this process failed to remove hydrophilic EDCs contaminants containing electron-withdrawing functional groups (diclofenac: 2.8%, DEET: 19.5%, carbamazepine:39%), thereby resulting in the accumulation of several persistent and hydrophobic EDCs in the sludge, which could pose a severe threat to environmental condition and ecosystem. Besides, longer SRT (180 days) is another limitation observed in this study.

Song et al. [310] reported the effects of salinity build-up on the performance of AnMBR regarding the removal of trace EDCs contaminants from municipal wastewater. Their findings revealed possible undesirable effects of increased salinity on the performance of the AnMBR reactor regarding the elimination of most hydrophilic EDCs contaminants. Moreover, higher removal of EDCs was achieved with no considerable impact of salinity accretion on the bioreactor. However, the potential application of this study might be restricted, due to comparatively low removal rates observed for hydrophobic compounds (such as atrazine, phenyl phenol, bisphenol A (BPA), and triclosan which could endanger the health of an aquatic and ecosystem.

#### 2.8.3. Optimization of MBRs Process in the Removal of EDCs from Water

Recent findings indicate that the efficacy of MBRs can be further optimized through the introduction of (i) superior filtration processes, for instance, NF, FO, and RO) and electrochemical separation efficient of removing micro-size EDCs contaminants [311,312], (ii) enhanced operating and cleaning approaches [313], and (iii) anaerobic processes, including electrogenesis and methanogenesis, resulting in a minimal energy requirement (and even energy recovery) [314,315], and strongly effectual degradation of contaminants [316].

Notably, bioaugmentation and prolonged HRT could enhance the removal efficiency of the EDCs contaminants. Kujawa-Roeleveld et al. [317] reported that some EDCs, such as ibuprofen (IBU), fenofibrate (FNF), and (acetylsalicylic acid (ASA), can be substantially degraded under anaerobic conditions and comparatively longer HRT (30 d). Similarly, about 81% removal efficiency and enhanced degradation (with 175 mg/L cephalosporin concentration) were attained when Saravanane and Sundararaman [318] operated an AnMBR system to remove cephalosporin derivative from pharmaceutical wastewater via bioaugmentation. The principal mechanism of removing EDCs contaminants during the sludge process has been biodegradation by microbes and sorption onto biomass.

Furthermore, in the quest to compliment the removal performance of the MBR process, Nguyen et al. [282] investigated the elimination of trace EDCs contaminants using hybrid-MBR processes with UV oxidation and high-pressure membrane processes (NF/RO). However, higher removals of all the steroid hormones, alkyl phenol-based surfactants, and industrial pollutants, and readily biodegradable hydrophilic EDCs contaminants were achieved. The integration of MBR with UV oxidation and NF/RO membrane technique led to a remarkable removal between (85–100%) and the concentration of EDCs were below the analytical detection limit. Findings also suggest that both UV oxidation and high-pressure membrane systems (RO/NF) could effectively augment the MBR process to enhance the removal of EDCs significantly. The authors also stated that MBR treatment is feasible for eliminating hydrophobic and easily degradable hydrophilic EDCs. However, some of the drawbacks of this process are low removal of carbamazepine either via MBR treatment or UV oxidation separately (17–32%), absorption and UV dispersion radiation as a result of influential existence of suspended solids in the influent, thus mitigating the general processes performance, unstable elimination of hydrophilic and persistent EDCs by MBR treatment and longer SRT for MBR (196 days).

Park et al. [319] reported that greater removal rate of target personal care and pharmaceutical compounds, such as acetaminophen, theophylline, caffeine, and naproxen (˃90%), tetracycline and mefenamic (˃80%) and atenolol, furosemide, ketoprofen (˃70%), was attained when two (PACL and chitosan) coagulants were added into the MBR system as compared to 17–23%) removals from control MBR. Moreover, from their study, improved membrane permeability and reduced irreversible membrane fouling were observed. Their findings concluded that higher removal efficiencies of target compounds in coagulation-MBR were achieved, due to enhanced biodegradation. However, both systems failed to remove carbamazepine and sulpiride compounds. Notably, the recalcitrant actions of carbamazepine and sulpiride could be due to input conjugate compounds transformed into the original compounds during the degradation [137,148]. Similar higher removals of pharmaceuticals and hormones EDCs were also reported in other studies [320,321,322]. Jiang et al. studied the effect of HRT on the removal of EDCs and fouling control in a hybrid moving bed biofilm MBR system and found that HRT (18 h) was the optimal condition for the highest removal rate of target EDCs pollutants, particularly ibuprofen (˃98.4%), salicylic acid (˃98.1%), primidone (˃90.9%), triclosan (˃90.6%), with lower removal of carbamazepine (˃26.2%) [323]. The authors concluded that more extended 18 h and 24 h HRT could substantially relegate membrane fouling.

Numerous studies have demonstrated that hydrophobic EDCs can be adsorbed to the biomass, leading to prolonged bioreactor retention time, thereby enhancing removal efficiency. However, non-biodegradable hydrophilic EDCs, such as diclofenac, fenoprop, carbamazepine, nonylphenoxyethoxyacetic acid and nonylphenoxyacetic acid, and metronidazole, were found to be persistent in both MBR and AS processes and are more recalcitrant, since they belong to electron-withdrawing groups, such as nitro groups (NO_2_), amide (CONH_2_), and carboxylic acid (COOH) [119,148,179,291,308,324].

In overview, the MBR technology has shown remarkable potential in removing EDCs contaminants. However, the inherent drawbacks associated with it are the inconsistency in the remediation performances and the uneconomical operational and maintenance costs involved. More so, the minimal TMP energy required by the MF is a desirable quality, but its lower retention is inadequate for portable sewage treatment owing to the low removals of EDCs. For better separation of the EDCs, NF and RO membrane may be recommended, but the optimal input energy and flux may be compromised, due to the accumulated EDCs (foulant). This shows that to achieve optimal use of this treatment method with minimal energy input and efficient removal of EDCs, more studies regarding membrane modifications are still required. Findings have revealed a critical need to consider the integrated system incorporating other recent water treatment technologies to treat these persistent contaminants effectively.

**Table 2 polymers-13-00392-t002:** Published information from various locations on different proportions and the negative influence of endocrine-disrupting compounds on the environment.

Matrices Type	Major Contaminants	Corresponding Concentrations (ng/L)	Locations	Major Effects	References
Drinking water	NP, BPA	505; 1430	France	Disruption of the normal hormone functions and physiological statue in human beings and animals.	[29]
Mangrove sediments	E2, EE2, E1	0.03–39.77; 0.45–129.78; 0.02–49.27 (ng/g)	Mangrove, Brazil	Feminization of fish, mimicking estrogens and disruption of homeostasis.	[322]
Biological wastewater treatment plants (WWTPs)	E1, E2, E2-17A, EE2	3050; 776; 2300; 3180	State of Ceará, Brazil	Negative effects on the reproductive and sexual systems in humans, wildlife, and fish.	[325]
River water	E1, E2, E3, EE2, BPA	26.5; 15.5; 3.6; 68.8; 37 (µg/L)	Malaysia	Diabetes, neurological challenges, cancer, tumors, obesity, damaged reproductive function, immune effects, heart disease in humans. Stimulation of breast cancer cells.	[326]
Surface water	E1, E2, EE2, E3	1.40–5.74; 1.10–5.39; 11.70–14.00; 2.15–5.20	Watershed, Istanbul, Turkey	Hazard to public health via uninterrupted exposure and food-chain.	[327]
Treated sewage effluent	AMP, SFZ, CBZ, SMZ, IBF	9299; 0.843; 125; 140; 203	Johor, Malaysia	Alteration of the potential function, the pattern of the male reproductive system, and male behavior.	[328]
Yellow river	E1, E2, E3, EE2, DES, EV, 4-t-OP, 4NP, BPA	2.98; 1.07; 4.37; 2.67; 2.52; 1.96; 89.52; 280.19; 710.65	China	Alterations in reproductive capacity and sex in aquatic organisms.	[329]
Tap water	NP, BPA	57.9 ± 18.0; 5.1 ± 8.8	Taipei and Kaohsiung (Northern and Southern Taiwan	Biomagnification and bioaccumulation through the food supply chain. Elevated-trophic-level species in humans and the ecosystem via food intake.	[330]
Surface water/sediment of the Yellow River	4-t-OP, BPA, E1, E2, TCS, 4-NP	4.7, 46.7; 1.3; ND; 6.8; 577.9	China	Uneven development of gonads and vitellogenin initiation in fish.	[331]
River water	BPA	˂215	Langat River Malaysia	Interrupt the release of adipokine that shield humans from a metabolic disorder.	[332]
Drinking water	BPA, NP, E1, E2	4.7−512; 8.2−918; ND-9.9; ND-3.2	China	Obesity, persistent miscarriages, and polycystic ovarian disease in women.	[333]
Drinking water	Styrene	45.11–203.48 (µg/L)	Johor Bahru Malaysia	Mimicking, blocking, development disturbances, and change function systems of hormones in humans and animals.	[334]
Surface sediments	BPA, 4-NP, 4t-OP	25.15; 356.5; 176 (µg/kg)	India	Irregularities in the reproductive system of aquatic species, humans and wildlife.	[335]
Surface water and fish muscle tissue	4-t-OP, NP, BPA, E1, E2, EE2, and E3	126.0; 634.8; 1573.1; 55.9; 23.9; 31.5; 5.2 and 26.4, 103.5, 146.9, 14.2, 9.3, 13.8, and 1.3 ng/g	Bahe River, China	Contamination in these regions triggered inhibition of gonad growth, elevated growth conditions, and repression of spermatogenesis in H. leucisculus.	[336]
Water, sediment, fish samples	OP, NP, BPA	ND-102; ND-127; ND ND-1.90 µg/g; ND-2.51 µg/g; ND-5.08 µg/g ND-643 ng/g; ND-476 ng/g; ND-1139 ng/g	Lagos Lagoon, Nigeria	It causes endocrine facilitated anomalies in fish, invertebrates, reptilian avian, and mammals (humans).	[337]
Estuarine water	DIC, E2, E1, EE2, Testosterone, Progesterone DMS, Propanolol, Caffeine, BPA	˂0.47–79.89; ˂5.28–31.43; ˂0.56–1.92; ˂ 0.30–7.67; 0.51–2.30; ˂0.41–0.46; ˂ 1.00–1.51; ˂ 0.25–0.34; 0.13–0.33; 0.19–0.47	Pulau kukup, Johor mariculture site, Malaysia	Influence on human development throughout the fetal period and potential cancer.	[338,339]
Mariculture fish	BPA, 4OP, 4NP	0.023; 0.084; 0.078 (ng/g)	Malaysia	Biomagnification and bioaccumulation via food web or food chain in humans.	[340,341]
Maternal blood and amniotic fluid	BPA, OP, TCS, NP	7.43 ng/mL and 7.75; 5.46 ng/mL and 5.72; 7.17 ng/mL and 7.04; 9.38 ng/mL and 8.44	India	These compounds have a propensity to cross the placental barrier and may affect the fetus.	[342,343]
River water and sediment	BPA	134; 275	Indonesia	An acute contaminant with severe impact on human organs, such as reproductive system, breast, adipose, and pancreas tissue.	[344]
Coastal groundwater/seawater	BPA, NP, E1	46.3–66.5 18.9–30.9	Coastal region of China	Disturb hormone biosynthesis and metabolism or cause a deviation from normal homeostatic control/reproduction.	[15,345]
Sewage water	Diethyl phthalate, estrone, nonylphenol-di-ethoxylate,	445–4635; 11–33; 747–3945	Hong Kong	Carcinogenicity, reproductive impairment, obesity and metabolic and disorders.	[346,347]

E1, estrone; CBZ, carbamazepine; EE2, 17α-ethinyl estradiol; E3, estriol; BPA, bisphenol A; NP, nonylphenol; OP, octyl-phenol; SFZ, sulfamethoxazole; E2, 17β-estradiol; DMP, dimethyl phthalate; 4tOP, 4-tert-octyl-phenol; SMZ, sulfamethoxazole; DMS, dexamethasone; DIC, diclofenac; DES, diethylstilbestrol; AMP, acetaminophen; EV, estradiol valerate; TCS, triclosan.

**Table 3 polymers-13-00392-t003:** Major membrane materials, fabrication techniques, modules, and nanomaterials used in membrane studies.

Major Membrane Materials	Fabrication Methods	Working Conditions	Pore Sizes (µm)	Nanomaterials Used	Membrane Configurations	Major Findings	References
PVDF	phase inversion	Nil	10.59–357	TiO_2_	HF	Enriched membrane strength and better hydrophilicity.	[21]
PES	Phase inversion	Pressure: 300 kPa	0.1–0.15	SiO_2_	HF	Expansion in adsorption site of the modified membrane.	[123]
PVDF	Irradiation graft polymerization and phase inversion	flow rates of 10 to 50 mL/min; Temp: 25 °C; Pressure: 0.02–0.2; MPa; pH: 6.5	0.00518	Functionalized polypropylene (PP) non-woven fabric	FS	Excellent mechanical properties and better reusability.	[266]
PVDF	NIPS	TMP: 0.21 Mpa, 0.14 Mpa, and 0.07 Mpa; pH: 7	NA	AgBiO_3_	FS	improving the flux, enhanced the protein fouling resistance. Resisted a bacterial fouling attack both in the presence and absence of visible light	[270]
PSF	Interfacial polymerization	Pressure: 5 bar; Temp: 25 °C	0.155	PA/TiO_2_	FS	Increased hydrophilicity.	[271]
PVDF	Phase inversion	Pressure: 0.1 bar; Temp: 24 °C; Filtration period: 10–50 m	0.25	CNT	FS	Substantially hydrophilic, enhanced wettability, additional open structured membrane developed.	[272]
PA	Electrospinning	Temp: 30 °C; s injection flow: 0.4 mL h^−1^; drum speed rate: 1000 rpm; voltage: 26 kV; working distance: 15 cm	0.02–0.09	Nanofibrous	HF	Considerably increased stability for the reuse application. Formation of homogenous fibers and porous structure with no significant changes during the reuse application. Excellent hydrophilic quality.	[273]
PVDF	Electrospinning	Temp: 27 °C; volume of permeates: 0.145–0.291; filtration period: 150 to 210 min; pressure: 1.0–25 bar; pH: 2.87 and 6.24	0.10501	MnO_2_	FS	Superior retention potential for the entire sampling time. Improved reusability for BPA removal.	[274]
CA	Diffusion induced phase separation	pH: 7; pressure: 200–1000 kPa	0.0003–0.001	Zeolites-zinc Oxide	FS	Introduction of Zeolite oxide build-up hydrophilicity of the membrane.	[275]
PSF	Phase inversion	Pressure: 200–1000 kPa	0.002–0.004	Polyaniline modified halloysite nanotubes (PANi-HNT) and	FS	A higher amount of PANi-HNT advances hydrophilicity and lead to a significant enhancement of the water flux	[276]
CA	NIPS separation (phase inversion)	T: 25 °C; contact time: 1–1500 m 250 rpm	50–200	Poly (4-vinylpyridine-b-ethylene oxide) (P4 VP-b PEO)	FS	Incorporation of 1% P4 VP-b-PEO to the membrane matrix enhances the adsorptive performance of the membrane.	[346]
PAN	Electrospinning	pH: 10.60; Time: 10 m; flow rate: 0.2 mL h^−1^	NA	Mono-porphyrin (2)	HF	Electrospinning supports shows excellent promise as a real-life application for water purification.	[347]

CP, 4-chlorophenol; PA, polyamide; HF, hollow fiber; FS, flat sheet; CNT, carbon nanotube; P4VP-b-PEO, Poly (4-vinyl pyridine-b-ethylene oxide); NIPS, non-solvent induced phase separation; NA, not available.

**Table 4 polymers-13-00392-t004:** Rejections of EDCs by membranes.

Major Contaminants	Treatment Process	Treatment Factors	Brief Procedure	Major Findings	Limitations	References
BPA, NP/WWTP influent	MF and RO membrane	Ph = 7.0–8.1; TOC (mg/L) = 0.8–77.4 TN (mg/L) = 0–52.5 EC (Ms/cm) = 0.1–3.1 UV (L/cm) = 0.00–1.77	Physicochemical water characterization. Analysis of micropollutant. Identification and quantification of compounds and filtration.	97% removal efficiency was achieved.	BPA (500 ng/L) was detected in the effluent. High energy demand.	[58]
BPA/biologically treated wastewater	MF and NF	Flux (MF) = 6.0–18.6 L/m^2^h; Flux (NF): 80 L/m^2^h; Temp = 21 °C TMP = 0.3 mPa (MF) 0.7 mPa (NF)	Circulation of the module with pure water. Determination of pure water infiltration.	Both techniques eliminate BPA. BPA removal efficiency: (61–75%) with NF.	Fouling. A decline in permeate flux in MF.	[74]
E1, E2, progesterone, testosterone/Purified water	UF membrane	MWCO: 1–100 kDa Pressure: 0.5–5 bar Pure water flux (L/m^2^h) 20.8–359.2 Final flux: 21.9–288.5 Time: 2–40 m Ph: 8	Stirring feed solution at 200 rpm for 16 h. Filtering of purified membrane for 30 min. Measurement of pure water flux. Collection of permeate.	Removal via solute-solute interactions for E1 corresponds to a higher proportion of organic matter at 25–50 mg/L for 10 kDa (48–52%); 100 kDa (33–38%) membranes.	Poor removals of E1 and hormone contaminants (52% and 38%).	[115]
BPA/drinking water	UF-PS (PS) membrane.	Temp: 25 ± 0.5 °C. pH: 7–13 BPA concentration: 100–500 μg/L. Ph: (3.68–10)	Measurement of pure water flux. Filtration of BPA solution.	Higher removal at the initial stage of the filtration.	Lower removal efficiency (20%). Fouling.	[117]
BPA, CBZ, IBF, and SFZ/drinking water	UF membrane	Operating speed: 50 psi. Flow rate: 0.65 L/m per cell.		Initial partial removal of BPA.	Poor BPA removal using modified PES membranes.	[118]
BPA, E2, E1, E3, EE2/synthetic wastewater	UF membrane	Working pressures (25, 30, 50, 75 kPa); temp: 20 ± 2 °C; TOC = 7 mg/L; pH: 7.6	Soaking of fresh membrane for 24 h. Removal of impurities. Determination of flux.	EDCs removal rates of (10–76%) were achieved via a fouled membrane.	Poor removal of E3 (10–17%).	[166]
BPA/model solution	NF and RO membranes	Temp: 45–50 °C; Maximum pressure: 31–83 bar, pH: 2–11; permeability: 0.85–4.86 (Lm^2^h) Filtration period:30–360 m		≥98% BPA rejection was achieved with polyamide-based RO membranes.	High energy demand. Too many modular units.	[167]
BPA and oxybenzone/feed solution	Nanocomposite membrane	Pressure: 1 bar; Filtration time: 2 h; Temp: 20 °C	5 mg/L BPA solution and 25 mg/L solution of oxybenzone was run via the HFM samples and the permeate was taken, and analyzed for oxybenzone/BPA using a UV–visible spectrophotometer	Higher removal of BPA (95%) and oxybenzone (98%) attained. Elevated pure water permeability (528.2 ± 44.6 mL/m^2^/h/mmHg)	Nil	[257]
BPA, DBP, DMP, DOP, NP/water	Nanocomposite membrane	Pressure: 0.02–0.5 Mpa; PEG feed concentration: 0.5 µg/L, operating pressure: 0.1 Mpa, Temp: 25 °C; pH: 6.5	The target contaminants were dissolved in deionized water. The filtration experiment was undertaken.	˃80% BPA removal was attained at 1.3 s contact time.	Nil	[268]
BPA/feed solution	UF(TFC) immobilized with TiO_2_		Preparation of feed solution. Quantification of the feed and the permeate solution.	Almost 99% BPA rejection was attained.	Nil	[271]
BPA/feed solution	Nanofibrous membrane	Temp: 30 °C, pH: 7; 150 rpm	Preparation of the stock solution. Batch experimentation. Sample analysis.	98% rejection of BPA was achieved.	Nil	[273]
BPA/drinking water	Nanocomposite membrane	Temp: 27 °C; Pressure ranges: 0.5–2.5 bars; filtration period: 1 h	The experiments comprise of uninterrupted filtration of BPA. The feed tank was loaded with 2 L of 500 ppb BPA. Collection of permeates and the solute for analysis.	Almost complete removal of BPA was achieved.	Nil	[274]
BP-3/feed solution	Nanocomposite membrane	pressure range: 200–1000 kPa; feed conc: 1, 3, 6 ppm;	Preparation of BP-3 solution. Measurement of the concentration of BP-3 in feed and permeate using UV spectrophotometer.	98% elimination of benzophenone-3 at pH (7).	Nil	[275]
Atrazine, oxybenzene/feed solution	Nanocomposite membrane	Pressure: 200 to 1000 kPa	The filtration unit is loaded with 3 ppm of EDCs and was run separately for each compound.	98% oxybenzene removal was attained.	Average removal of atrazine (50%)	[276]
BPA/feed solution	UF membrane	pH: (3–13) MWCO: 100 Da TMP: 0.1 × 10^6^–0.3 × 10^6^ Pa Temp: 20 ± 2 °C BPA conc.: 5 mg/L	The UF membrane was installed, and the solution was introduced into the UF cup. Magnetic stirring.	Both salt and acidic Ph improve the transportation of BPA.	Decline BPA rejection decreased significantly when the BPA molecule was ionized.	[348]
DMP, DEP, DBP, DnOP, DEHP/water	NF membrane	pH: 4–9; pure water flux: 47.5 L/m^2^h; Temp: 25–45 °C.	Preparation of a feed solution. Measurement of concentrations of PAEs in both the feed and permeate.	Removal efficiencies of 95.4%; 95.1% and 91.5% were recorded for DEHP, DnOP, and DBP.	Lower adsorption rates. Low rejection of sulfamides.	[349]

E1, estrone; IBF, ibuprofen; E2, 17β-estradiol; SFZ, sulfamethoxazole; EE2, 17α-ethinyl estradiol; PPCPs, pharmaceutical personal care products; BPA, bisphenol A; NP, nonylphenol; TCS, triclosan; E3, estriol; CBZ, carbamazepine; MF, microfiltration; UF, ultrafiltration; NF, nanofiltration; RO, reverse osmosis; PAEs, phthalate acid ester; DMP, dimethyl phthalate; DBP, dibutyl phthalate; DEP, diethyl phthalate; DBP, dibutyl phthalate; DnOP, di-n-octyl phthalate; DEHP, diethylhexyl phthalate; DOP, dioctyl phthalate; UPW, ultra-pure water; BP-3, benzophenone-3.

**Table 5 polymers-13-00392-t005:** Removals of some endocrine-disrupting compounds during MBR processes.

Major Contaminants/Sources	Treatment Process	Treatment Factor	Brief Procedure	Major Findings	Limitations	References
Steroid hormones, alkyl phenolic surfactants, pesticides, PPCPs, and industrial chemicals/synthetic wastewater	MBR treatment with UV oxidation/(NF and RO)	pH. (7.2–7.5): initial start-up and acclimatization (51 d), MBR period (5 d).	Preparation of stock solution. Introduction of stock solution to the synthetic Wastewater.	Acetaminophen removal = 87.1% carbamazepine removal ≥ 96%.	Removal efficiency by MBR and UV varied from 17 to 32%. Long term operation of MBR (196 days).	[284]
Steroidal hormones, xenoestrogens, pesticides, caffeine, pharmaceuticals, and personal care products (PPCPs)/wastewater	AMBR	SRT: 10–15 d; HRT; 1 d; MLSS: 7.5–8.5 g/L; aeration: 10 m cycle; pH: 7.0–7.7.	Loading of the influent into the reactor. Supply of compressed air to the aerobic tank. Sample collection and analysis	(˃90%) removals of target trace organic contaminants	(24–68%) removals for DIC, CBZ, amitriptyline, trimethoprim, diazepam, gemfibrozil, omeprazole, sulfamethoxazole and fluoxetine	[304]
BPA/ samples from sewage plants	secondary treatment and MBR.	HRT = 20–180 0.3–2.8, 1.1–7, 5.3–25 SRT (d) = 3.3–20 MLSS = 780–6900 (mg/L) Temp (°C): 3–25, 2–23 6–23, 7–24.	ND	Removal efficiency ranged between 1–77%	Chemically assisted primary treatment achieved very low removal.	[306]
BPA/Synthetic municipal wastewater	MBR, OMBR, FO membrane	HRT:4,8,12 h) MLSS: 6.5 g/L and 8.5 g/L BPA sludge loading: (0.05–0.16 mg/g/d)	ND	Overall removal of 70% was achieved. Conventional MBR: 93.9% and 98%, respectively.	The rejection of BPA by the MF membrane was very low (10.3%). Salt leakage problem. Fouling.	[307]
Caffeine, naproxen, and acetaminophen/wastewater	AMBR	average flow rate: 8800 m^3^/d; Temp: 21 °C; HRT: 11 h; SRT: 6–8 d, MLSS: 5700 mg/L; daily sludge production: 90 m^3^/d.	Feeding the reactor with an influent sample. Continuous aeration at 2.5 mg/L. Sample collection and analysis.	Complete removal of target compounds.	Negative removals of trimethoprim (−2%) and clarithromycin (−34%)	[308]
BPA, TCS, diazinon, triclocarbon, 4-n-nonylphenol, caffeine, DIC, CBZ/synthetic wastewater	Anaerobic membrane bioreactor (AnMBR)	Digester temp: (35 ± 1 °C), HRT: (4 d), permeate flux: (1.8 L/m^2^ h) and organic loading rate: (1.3 gCOD/L d), MLSS concentration: (10 g/L), SRT: (180 d).	Inoculation. Dilution of wastewater. Acclimatization. Analysis of sludge and permeate samples.	(˃70%) hydrophobic contaminants were removed. ˃70% of hydrophilic EDCs were removed.	Poor removal of diclofenac (2.8%), DEET (19.5%) and carbamazepine (39.2%), respectively.	[309]
Clozapine, butylparaben, diazinon, triclocarbon, NP, atrazine (herbicide), phenyl phenol, BPA, and TCS/municipal wastewater	Biosorption and biodegradation	HRT = 5 d; mixed liquor pH = 7 ± 0.1; temp = 35 ± 1 °C	Feeding the Bioreactor. Circulation of digested sludge. Mixing of the sludge.	Trimethoprim, carazolol, hydroxyzine, amitriptyline, and linuron removals = ˃80%. The removal of phenyl phenol was 60%.	Poor Atrazine removal 6.8%. BPA removal ranged between 40% to 20%.	[310]
TCT, CBZ, DIC, caffeine, theophylline, naproxen, acetaminophen, mefenamic, atenolol, furosemide, ketoprofen/wastewater	Coagulated-AMBR	HRT: 9 h; SRT 25 d; MLSS: 8 g/L; pH: 6–8	Loading of the influent into the reactor. Supply of compressed air to the aerobic tank. Sample collection and analysis.	Acetaminophen, theophylline, caffeine and naproxen (˃90%); TCT, mefenamic (˃80%) removals,	CBZ, sulpiride and DIC	[319]
DIC, IBF, EI, EE2/synthetic wastewater	AASMBR	T: 8 °C and 12 °C; SRT: 30, 60, 90 d; Time points: 0.2, 0.5, 1.5, 2.3, 3, 4.5, 5, 8.5, 12, 12.5, 24 h; flow rate: 15 L d^−1^; operational flux: 0.14 md^−1^. Fine	Instantaneous spiking of EDCs stock solutions to the reactors. Constant aeration. Sample analysis (LS-MS/MS).	Complete removal of IBF and E1, EE2 (66%).	Poor removal of DIC (31%).	[320]
BPA, DIC, CBZ, BIS/urban wastewater	AMBR	Temp: 7–20 °C; HRT: 35 h; aeration intensity: 0.4–0.6 m^−3^m^−2^h^−1^; pH: 6.6–7.3; operation interval: 120 d; Flux (continuous): 7.8 Lm^−2^h^−1^.	The MBR unit was loaded with clarified wastewater. The SRT was maintained via sludge stabilization in the membrane compartment. Collection of treated permeate water and wasted sludge in the permeate and sludge tanks.	BPA (97%), Bisoprolol (65%)	Membrane fouling. Poor removal of CBZ (28%) and DIC (38%).	[350]

DIC, diclofenac; E1, estrone; CBZ, carbamazepine; E2, 17β-estradiol; BIS, bisoprolol; NP, nonylphenol; EE2, 17α-ethinyl estradiol; PPCPs, pharmaceutical, and personal care products; BPA, bisphenol A; TCS, triclosan; DIC, diclofenac; TCT, tetracycline; HRT, hydraulic retention time; NF, nanofiltration; SRT, solid retention time; AnMBR, anaerobic membrane bioreactor; AMBR, aerobic membrane bioreactor; MLSS, mixed liquor suspended solids; TMP, transmembrane pressure; SRT, solid retention time; TSS, total suspended solids; ND, not disclosed.

## 3. Conclusions

The efficient treatment of potable water and wastewater to eliminate emerging and persistent EDCs microcontaminants confronts several complicated challenges that require novel, sustainable, eco-friendly solutions.

Several studies have reported that conventional treatment technique is inadequate to remove EDCs microcontaminants from water, since the contaminants are still in abundance in the effluent discharge from the system. Lack of regional and global stringent discharge limits for these recalcitrant micropollutants, despite their adverse effects on intact organisms and ecosystems even at minuscule proportions, indicated that the regulatory authorities have not adequately combat the issue at the present moment. Consequently, the presence of EDCs in the effluent from the conventional treatment process is expected to become a critical issue in the near future owing to the rapid industrial advancement and expeditious growing population. Hence, justifying continuous research and monitoring of the concentrations of the emerging contaminants in drinking water. Notably, there is a limited dedicated review on the recent trend in removing EDCs from water using membrane and MBR technologies.

This review paper summarized and discussed various research findings on removing endocrine-disrupting compounds via membranes and MBR techniques. A superficial insight into the content of this review indicates that the essential mechanisms controlling the removal of endocrine-disrupting compounds from water using membrane techniques via adsorption, steric hindrance, and electrostatic interactions are adequately discussed.

Similarly, the removal mechanisms, such as biodegradation and sorption, dictating the removal process of EDCs via MBR systems, are succinctly reviewed. The knowledge of these mechanisms involved in the rejection process is an essential underlying strategy to influence and enhance the efficacy of the membrane and MBR procedure. From the above review, it is evident that the utilization of membrane-based technologies has proven to be an excellent approach for eliminating EDCs from potable water. This is because membrane technologies, apart from being physical separation processes, exhibit facile operation, sustainability, high efficiency, cost-effectiveness, and broader potential for extensive scale application in water treatment.

This paper also clearly revealed that membrane surface modification via nanoparticles involving the addition of hydrophilic functional groups can be considered one of the most suitable and promising panacea options to extenuate fouling challenges, and simultaneously, enhance the membrane properties, including hydrophilicity, permeability, and reusability without compromising the thermal and mechanical properties of the membrane mainstay. Finally, there is a need for further studies to exploit various nanoparticles and semiconductors to modify the membrane to enhance its hydrophilicity properties and EDCs removal efficiency. Additional investigations on MBR efficiencies at longer SRTs and HRTs may need to be considered for compounds that exhibit partial sorption and degradation/transformation. Continuous monitoring of EDCs proportions in the environment is strongly recommended. The integration of membrane and other phase-changing technologies and advanced oxidation process (AOPs) techniques into a single system could harmonize each other to conquer the challenges from both systems and led to a highly efficient potable water and wastewater treatment process.

## Figures and Tables

**Figure 1 polymers-13-00392-f001:**
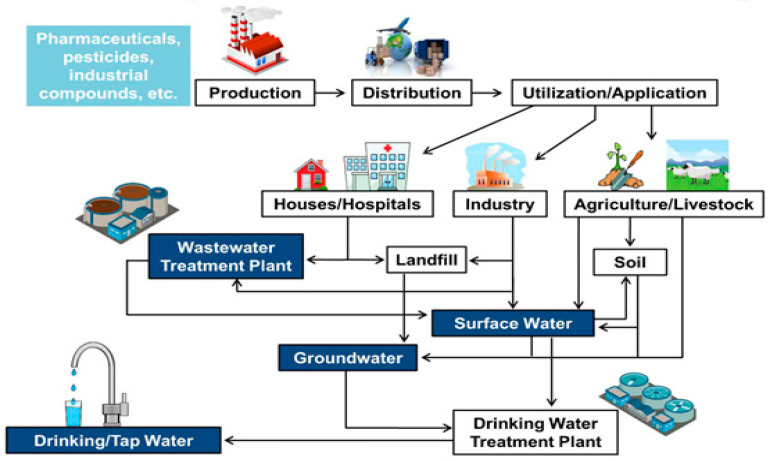
Typical routes and sources of endocrine-disrupting (EDCs) contaminants in the environment [43].

**Figure 2 polymers-13-00392-f002:**
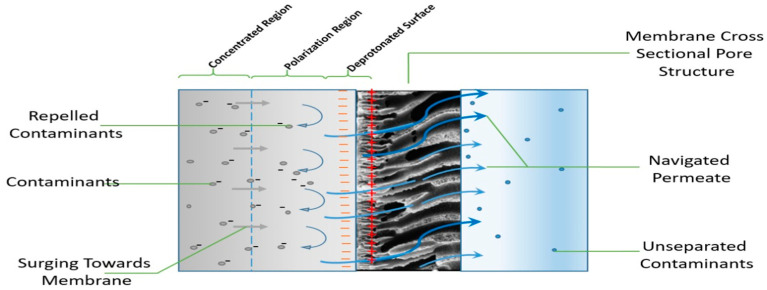
Removal mechanisms for EDCs during membrane processes via adsorption, sieving mechanism, and electrostatic interactions.

**Figure 3 polymers-13-00392-f003:**
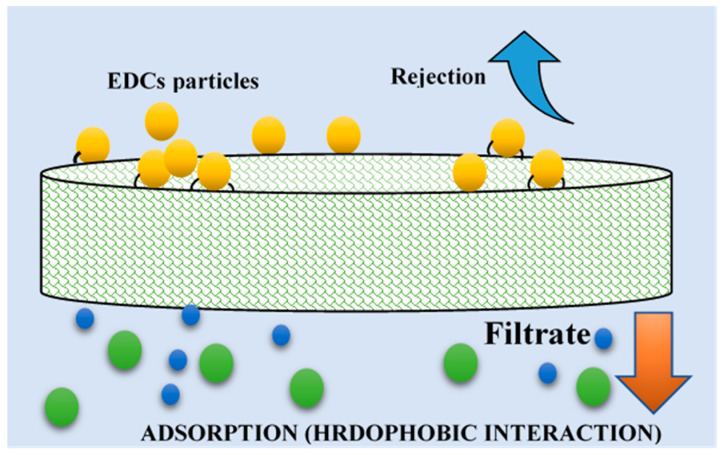
Adsorption (hydrophobic interaction) adapted with modification [128].

**Figure 4 polymers-13-00392-f004:**
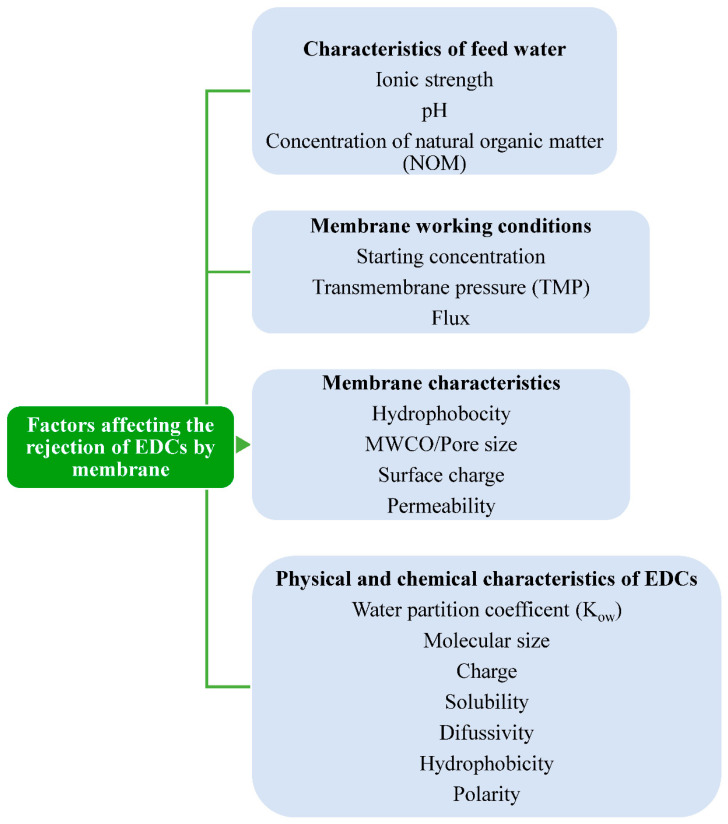
Essential Factors that influence EDCs removal during membrane process. Adapted with modification [123].

**Figure 5 polymers-13-00392-f005:**
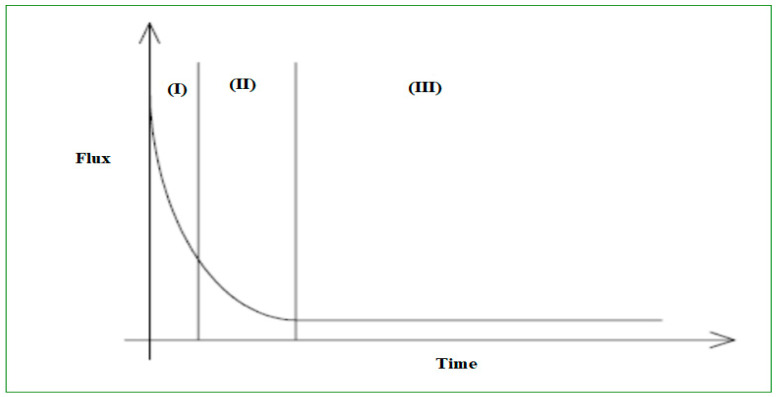
A Schematic representation of the stages in flux decline [218].

**Figure 6 polymers-13-00392-f006:**
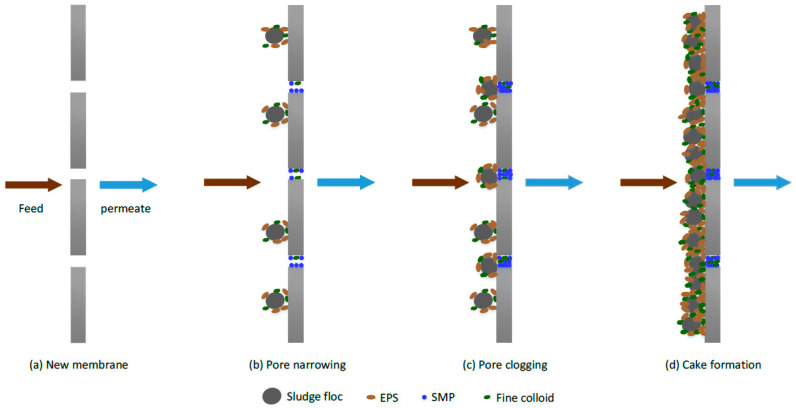
Schematic representation of membrane fouling mechanisms in MBRs showing different stages of fouling, (**a**) New membrane; (**b**) Pore narrowing; (**c**) Pore clogging; (**d**) Cake formation [202].

**Figure 7 polymers-13-00392-f007:**
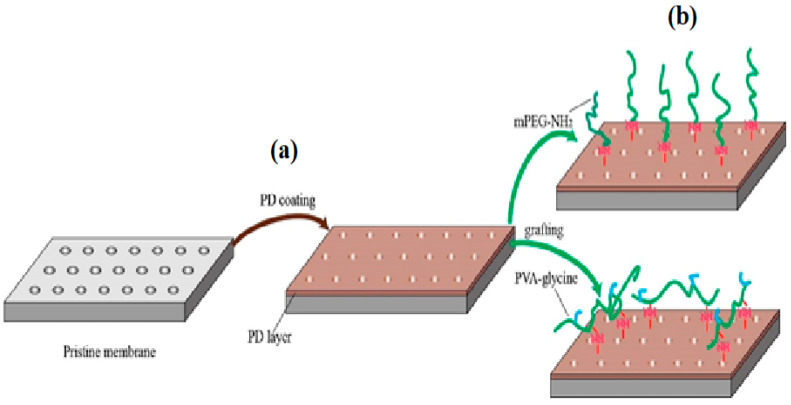
Schematic diagram of the surface modification technique (**a**) surface coating and (**b**) surface grafting adapted from [206].

**Table 1 polymers-13-00392-t001:** Outline of membrane processes and their characteristics in order of decreasing nominal pore sizes [91,94,95].

Membranes	MF Symmetric/Asymmetric	UF Asymmetric	NF Asymmetric	RO Asymmetric
Pore size	0.025–5 (µm)	1–100 nm	0.5–10 nm	˂1 nm
Thickness (µm)	10–150	150–250	150	150
Operating pressure (bar)	0.1–10	0.1–10	10–50	35–170
Flux range (Lm^−2^h^−1^bar^−1^)	˃50	10–50	1.4–12	0.05–1.4
Separation mechanism	Sieving	Sieving	Sieving and electrostatic	Solution diffusion
Applications	Clarification Pre-treatment Removal of bacteria	Removal of macromolecules, bacteria, viruses.	Removal of (multivalent) ions and relatively small organics.	Ultra-pure water. Desalination.
Rejection:				
Monovalent ions	-	-	-	+
Multivalent ions	-	−/+	+	+
Small organic compounds	-	-	−/+	+
macromolecules	-	+	+	+
Particles	+	+	+	+

MF, microfiltration; UF, ultrafiltration; NF, nanofiltration; RO, reverse osmosis.

## Data Availability

Not applicable.

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
