# Peer review of "Recent Advances in the Rejection of Endocrine-Disrupting Compounds from Water Using Membrane and Membrane Bioreactor Technologies: A Review"

_polymers, 2021, doi:10.3390/polym13030392_

Round 1
Reviewer 1 Report
The manuscript titled "Recent advances in the rejection of endocrine-2 disrupting compounds from water using membrane 3 and membrane bioreactor technologies: A Review" by Katibi et al. is an interesting work. It has potential for providing key insights to the membrane development technology for edc removal. The work can be further benefited with the following suggestions, I believe-
- Introduction to electrospun/solution blow spun nanofiber based membranes for EDC removal in this article, given the said technologies are coming up as quite efficient ones.
- Some figures like Figure 3 etc. seemed to have been missing references, if they are taken from other sources and are modified.
- The anti-fouling section concerning the membranes should come in conclusion as a authors' personal views, because the section is too small and doesn't have much references.
- In many cases authors have merely stated the fact about membranes' involvement in removing EDCs, like lines 409-412. Authors should revise the manuscript to elaborate the underlying mechanisms in those cases, to show their authority in this domain.
- In several places there are text formatting issues likes of line 393, or improper subscript, like- line 366 or figures in general, which seem stretched, and unclear. Also the line spacings are different in different places of article.
Authors need to revise the article to provide more of their inputs instead of just filling it with published work and repeating same concepts in different sections. Like authors described same NF, UF etc. in different areas multiple time. Either they should make them as sections with physical separation techniques and electrically assisted techniques or based on NF and UF techniques.
Author Response
Good day sir/ma,
Attached herein is the Authors response to reviewer's comments.
Thank you.
Regards.

Reviewer 2 Report
This paper reviews the recent advances in the rejection of endocrine-disrupting compounds from water using membrane and membrane bioreactor technologies. Occurrences of EDCs and their potential environmental risks were first presented, followed by the removal using membrane filtration and MBR processes. In general, this paper synthesizes a lot of literature regarding the EDCs and their removal by membrane-based processes. However, before it can be accepted for publication, the following points need to be addressed.
(1) Fig. 5 is not impressive. No much information is shown in this figure. Similarly, Fig. 6 is also not very meaningful. Description in the main text might be a better way.
(2) Section 2.8 is too simple. Only one sub-section ---Mechanisms of EDCs removal during MBR process is shown. In fact, there are many topics regarding EDCs removal in MBRs. This subsection can be broken into several sections, including mechanisms, performance, process optimization and so on.
(3) Also, there is already several review paper regarding the removal of micropollutants using MBRs, such as publications in Bioresource Technology. Please refer to the available literature to further polish the corresponding section in the current paper.
(4) Section 1.2--Adverse Effects of EDCs on the environment is not very clear. Comprehensive summary of the Adverse Effects of EDCs should be conducted rather than describing their occurrence, concentration and so on in many paragraphs.
(5) Section 2.2--- Removal mechanisms of Endocrine-disrupting compounds during membrane processes. A better organization is needed, for instance, the mechanisms can be documented based on membrane types, such as MF/UF, and NF/RO.
(6) Section 2.3 --- Factors affecting the membrane rejection performance of the EDC. Similarly, I suggest that the discussion on factors should be organized based on different types of membranes,
(7) Section 2.4 -- Membrane Fouling challenges is quite simple. More information regarding this topic should be provided.
(8) Section 2.5--- Membrane Fouling Extenuation Techniques is also quite simple. One possible way is to remove it, and the other is to merge this section with section 2.4.
(9) There is no section 2.7. However, section 2.7.2--Nanomaterials and membrane technology is presented.
(10) More informative figures summarizing the key data or mechanisms should be provided. In its current version, the information presented in this figure seems simple.
Author Response
Good day sir/ma,
Attached herein is the author's response to the reviewer's comment.
Thank you.
Best regards.
